# 2.7 Å cryo-EM structure of human telomerase H/ACA ribonucleoprotein

George E. Ghanim [1,2], Zala Sekne[1,2], Sebastian Balch[1],
Anne-Marie M. van Roon[1] & Thi Hoang Duong Nguyen [1] ✉

Telomerase is a ribonucleoprotein (RNP) enzyme that extends telomeric repeats at eukaryotic chromosome ends to counterbalance telomere loss caused by incomplete genome replication. Human telomerase is comprised of two distinct functional lobes tethered by telomerase RNA (hTR): a catalytic core, responsible for DNA extension; and a Hinge and ACA (H/ACA) box RNP, responsible for telomerase biogenesis. H/ACA RNPs also have a general role in pseudouridylation of spliceosomal and ribosomal RNAs, which is critical for the biogenesis of the spliceosome and ribosome. Much of our structural understanding of eukaryotic H/ACA RNPs comes from structures of the human telomerase H/ACA RNP. Here we report a 2.7 Å cryo-electron microscopy structure of the telomerase H/ACA RNP. The significant improvement in resolution over previous 3.3 Å to 8.2 Å structures allows us to uncover new molecular interactions within the H/ACA RNP. Many disease mutations are mapped to these interaction sites. The structure also reveals unprecedented insights into a region critical for pseudouridylation in canonical H/ACA RNPs. Together, our work advances understanding of telomerase-related disease mutations and the mechanism of pseudouridylation by eukaryotic H/ACA RNPs.

Telomeres are nucleoprotein structures that protect the ends of linear eukaryotic chromosomes[1]. Despite the importance of telomeres for long-term proliferative capacity, they shorten with each cellular division due to incomplete genome replication at linear DNA ends[2]. Short telomeres signal replicative senescence, resulting in genome instability and cell death[3–6]. Telomerase counteracts this telomere loss by de novo RNA-templated DNA synthesis of telomeric repeats[7]. The telomerase reverse transcriptase subunit (TERT) copies telomeric repeats using an internal template sequence contained within a larger telomerase RNA (hTR in humans)[7]. Telomerase activity is upregulated in stem cells, germline and cancer cells, whereas telomerase deficiencies result in a range of premature aging diseases[8].

Human telomerase adopts a bilobed architecture, in which its 12 protein subunits are tethered by a flexible hTR scaffold[9,10]. The lobe that carries out DNA synthesis of telomeric repeats is called the catalytic core. In the catalytic core, the pseudoknot/template (PK/t) domain and conserved regions 4 and 5 (CR4/5) of hTR associate with

TERT and a histone H2A-H2B dimer (Fig. 1a, left, c)[10,11]. The other lobe, called the Hinge and ACA (H/ACA) RNP lobe, is crucial for telomerase biogenesis. The H/ACA RNP lobe is built upon the H/ACA domain of hTR, which adopts a double-hairpin architecture (Fig. 1a, right). Each hairpin is bound by a heterotetramer of the core H/ACA proteins (dyskerin, NOP10, NHP2 and GAR1) (Fig. 1b, c)[12,13]. Furthermore, the telomerase H/ACA RNP also contains a Cajal body localization factor, TCAB1 (Fig. 1b, c)[14,15]. Numerous mutations in the H/ACA subunits have been identified in patients with telomere disorders including dyskeratosis congenita and Hoyeraal-Hreidarsson (HH) syndrome[16,17]. The affected patients share telomerase deficiency and premature aging as common phenotypes, underscoring the importance of the telomerase H/ACA RNP lobe in telomere maintenance[16].

Besides telomere maintenance, members of the H/ACA RNP family, such as small nucleolar RNPs (snoRNPs) or small Cajal body-specific RNPs (scaRNPs), have a general role in the pseudouridylation of ribosomal and spliceosomal RNAs[18–21]. Pseudouridine is the most

[1]MRC Laboratory of Molecular Biology, Cambridge CB2 0QH, UK. [2]These authors contributed equally: George E. Ghanim, Zala Sekne.
✉e-mail: knguyen@mrc-lmb.cam.ac.uk

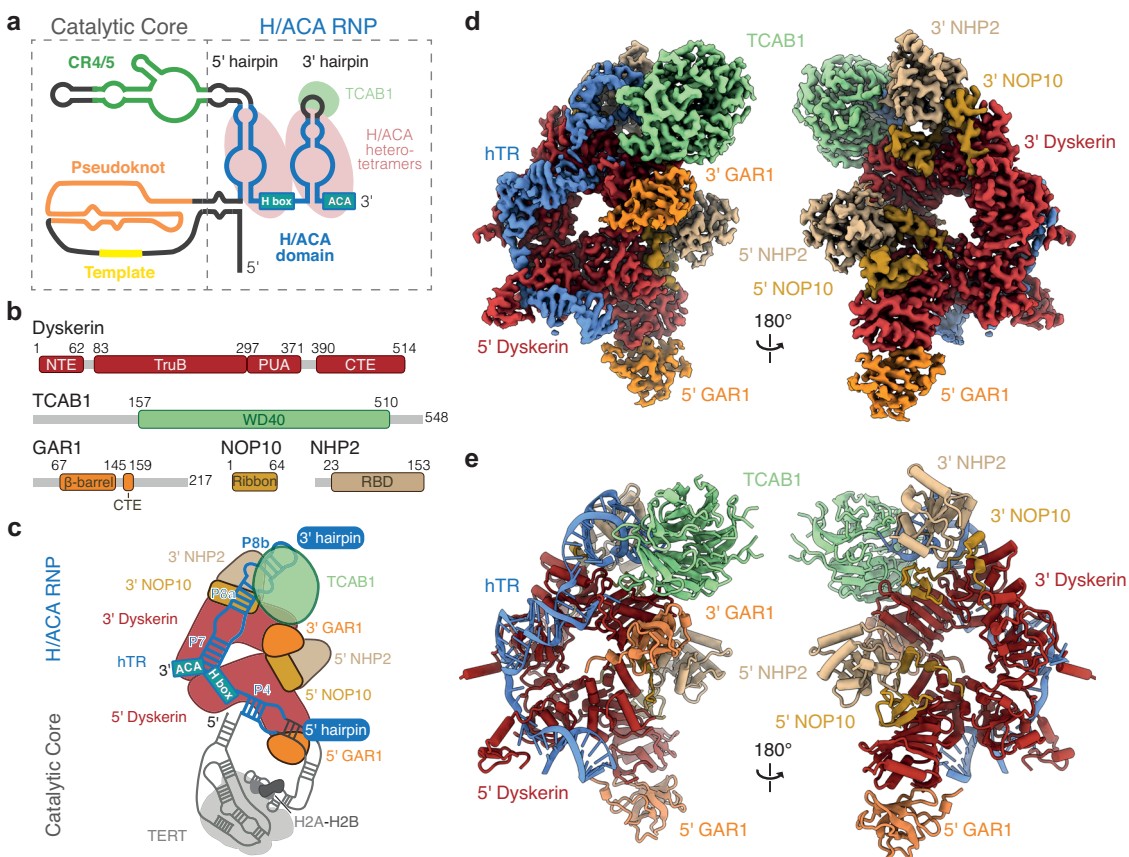

**Fig. 1 | Structure of human telomerase H/ACA RNP. a** Secondary structure of hTR. RNA domains (colored as indicated) are divided into the catalytic core and the H/ACA RNP. The binding of the H/ACA heterotetramers and TCAB1 on the 5' and 3' RNA hairpins is shown. **b** Domain architectures of H/ACA protein subunits. NTE, N-terminal extension; TruB, tRNA pseudouridine synthase B-like; PUA, pseudouridine synthase and achaeosine transglycosylase; CTE, C-terminal extension; and RBD, RNA-binding domain. **c** Schematic of the human telomerase holoenzyme. This work focuses on the H/ACA RNP, thus the catalytic core is greyed out. **d** 2.7 Å cryo-EM reconstruction of the telomerase H/ACA RNP (Supplementary Figs. 3, 4). The subunits are colored as in (**b**). **e** A ribbon representation of the telomerase H/ACA RNP with subunits colored as in (**b**).

abundant RNA post-transcriptional modification in cells and is essential for the biogenesis and function of the ribosome and spliceosome[18–22]. The catalytic pseudouridine synthase subunit of H/ACA RNPs (Cbf5 in archaea and yeast, and dyskerin in human) converts a substrate uridine to a pseudouridine (Ψ)[22–24]. Within the H/ACA RNA, large internal loops, called pseudouridylation pockets, confer substrate specificity by base-pairing with the RNA substrate on either side of an unpaired "UN" dinucleotide (U is the target and N is any RNA base)[19,25–28]. Based on single-hairpin archaeal structures, a region within Cbf5 (dyskerin orthologue), called the thumb loop, influences pseudouridylation catalysis by adopting different conformations[27–33]. However, the entire dyskerin thumb loop has never been visualized in a eukaryotic H/ACA RNP. It is also unknown whether the eukaryotic thumb loop adopts similar conformations as the archaeal counterpart and how this influences the pseudouridylation active site of dyskerin.

Structural information on eukaryotic double-hairpin H/ACA RNPs has come solely from recent cryo-electron microscopy (cryo-EM) studies of the human telomerase holoenzyme resolved at 3.3–8.2 Å resolution[9,10,34,35]. The previous structures[9,10,34,35] defined the architecture of double-hairpin H/ACA RNPs and mapped interactions within the H/ACA RNP. Here we present a 2.7 Å cryo-EM structure of the telomerase H/ACA RNP, the highest resolution achieved for telomerase cryo-EM maps to-date. Our work explains the molecular pathology of numerous unmapped disease mutations and captures conformational dynamics of the dyskerin thumb loop, providing important insights into eukaryotic pseudouridylation.

## Results

### Structure determination of the telomerase H/ACA RNP lobe

We previously collected a cryo-EM dataset of 41,053 micrographs for the DNA-bound telomerase in complex with recruitment factors TPP1 and POT1, and resolved the structures of the TPP1-POT1 bound catalytic core of telomerase to 3.2–3.9 Å resolution[36]. We reasoned this large dataset could also offer improved resolution and aid interpretability of the H/ACA RNP lobe over our previous 3.4 Å map[10]. By combining signal subtraction[37] with extensive 3D classification (see Methods), we obtained a reconstruction of telomerase H/ACA RNP at 2.7 Å overall resolution with a local resolution range of 2.7–5.3 Å (Fig. 1d, e, Supplementary Figs. 1, 2, Supplementary Table 1, and Supplementary Data 1). This significantly improved resolution allowed high-confidence modeling for all H/ACA protein subunits and hTR (Supplementary Figs. 3, 4). We resolved numerous regions that were previously unresolved due to conformational heterogeneity, yielding the most complete atomic model for a eukaryotic H/ACA RNP (Supplementary Fig. 5, Supplementary Table 2, and Supplementary Data 1)[10]. To gain insight into the mechanism of the general H/ACA RNP pseudouridylation, we performed 3D variability analysis and focused classification on the thumb loops of the two dyskerin subunits. We resolved two different conformations of the thumb loop of the 3' dyskerin subunit.

### Additional interactions are identified between the dyskerin N-termini

The telomerase H/ACA RNP contains two H/ACA heterotetramers of dyskerin, NOP10, NHP2, and GAR1, each assembled on the 5' and 3'

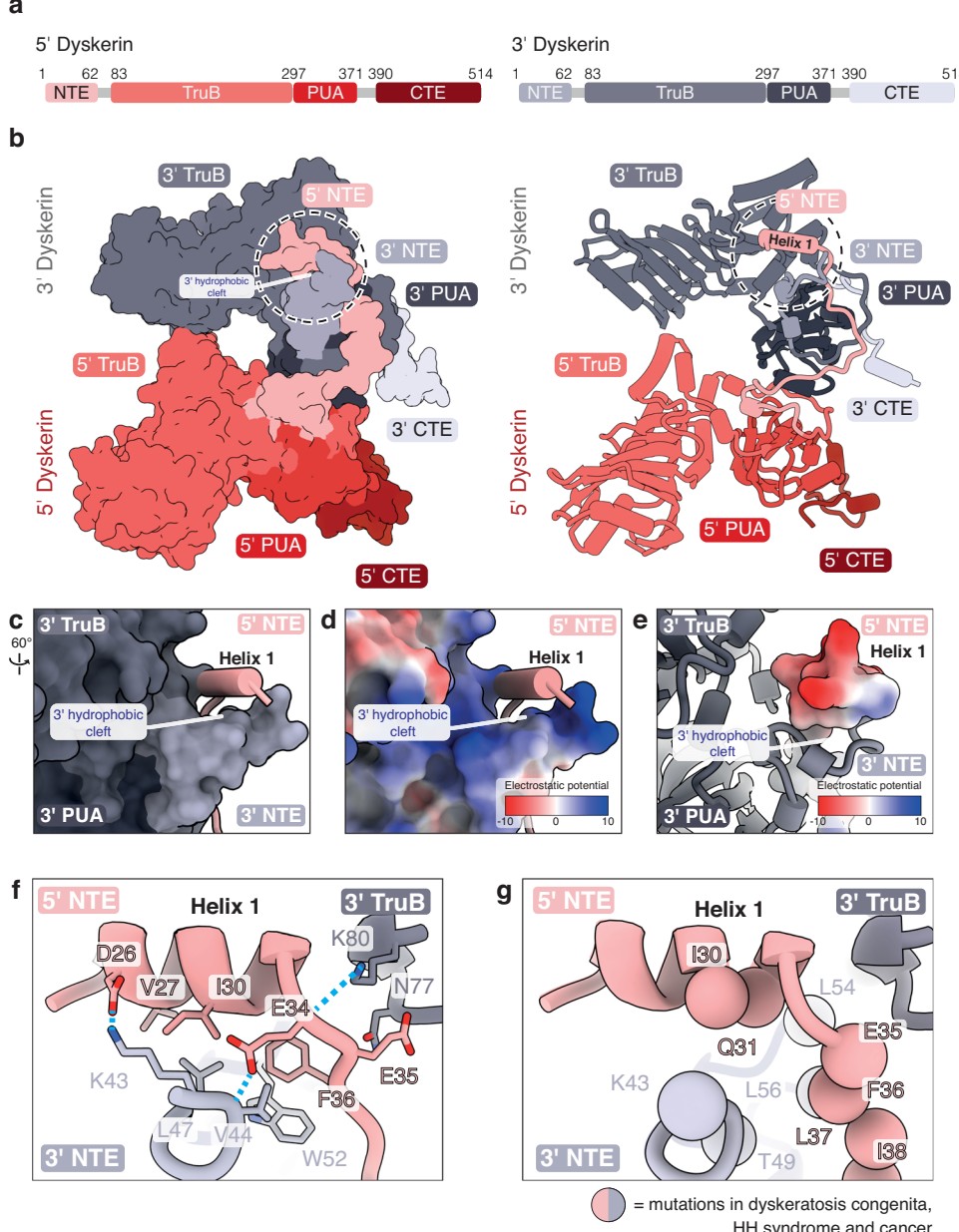

**Fig. 2 | Cross dyskerin-dyskerin interactions. a** Domain architectures of the 5′ and 3′ dyskerin. The domain color scheme is used throughout this figure. **b** Structure of the 5′ and 3′ dyskerin in the telomerase H/ACA RNP in space-filling (left) and ribbon (right) representations. The 3′ hydrophobic cleft and helix 1 of the 5′ dyskerin are labeled. For clarity, only domains of the dyskerin subunits are shown. **c** Interactions between helix 1 of the 5′ dyskerin and the 3′ hydrophobic cleft formed by the 3′ dyskerin. **d, e** The electrostatic potential of the 3′ hydrophobic cleft formed by the 3′ dyskerin and helix 1 of the 5′ dyskerin, respectively. **f** Cross-dyskerin interaction at the 3′ hydrophobic cleft (also see Supplementary Fig. 4b). Hydrogen-bonds are shown as dashed blue lines. **g** Disease mutations at the inter-dyskerin interface shown in (**f**). Residues whose mutations are associated with dyskeratosis congenita, HH syndrome and sporadic cancer are highlighted as spheres.

RNA hairpins of hTR (Fig. 1c)[9]. Henceforth, we refer to each hetero-tetramer and its constituent subunits as either 5′ or 3′ accordingly. The two H/ACA heterotetramers extensively interact with one another, mainly via the two dyskerin subunits. Dyskerin consists of four domains: an N-terminal extension (NTE), a pseudouridine synthase and achaeosine transglycosylase (PUA), a tRNA pseudouridine synthase B-like (TruB) and a C-terminal extension (CTE) domain (Fig. 2a). The NTE, PUA and TruB domains of each dyskerin molecule form a platform for the other dyskerin subunit to bind (Fig. 2b)[10]. Although the inter-dyskerin interfaces have been shown to be a hotspot for disease mutations, many mutations were unresolved in previous structures and thus not accounted for[9,10,34,35].

The improved map enabled us to not only model regions of both dyskerin NTEs, which harbor unresolved disease mutations, but also discover new inter-dyskerin interactions (Fig. 2b, circle, c, f, Supplementary Figs. 4b, 5 and Supplementary Table 2)[10,32]. The newly modeled region of the 5′ NTE (residues 24-33) forms an α-helix (helix 1) (Fig. 2b, right circle, c; see also Supplementary Data 1). On the other hand, the newly modeled region of the 3′ NTE (residues 43-47) forms a hydrophobic cleft with the rest of the 3′ dyskerin NTE and part of the 3′ dyskerin TruB domain (Fig. 2c–f). The 3′ hydrophobic cleft accommodates helix 1 of the 5′ dyskerin via numerous hydrophobic interactions between the 3′ dyskerin (V44, L47 and W52) and helix 1 of the 5′ dyskerin (V27, I30 and the highly conserved F36) (Fig. 2c, f). Helix 1

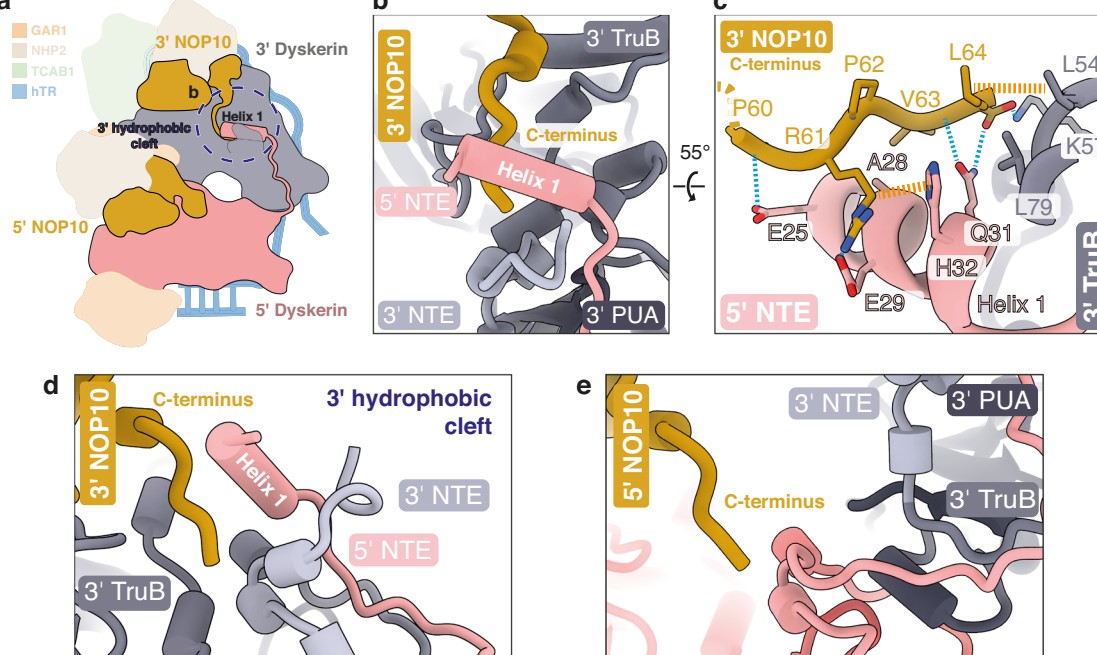

**Fig. 3 | Enhancement of the inter-dyskerin interaction by the 3' NOP10.**
**a** Schematic of the telomerase H/ACA RNP. The 5' and 3' dyskerin and the 5' and 3' NOP10 are highlighted. The 3' hydrophobic cleft is circled with a dotted line.
**b** Close-up view of helix 1 of the 5' dyskerin, the NTE, TruB and PUA domains of the 3' dyskerin, and the C-terminus of the 3' NOP10. **c** Interactions of the 3' NOP10

C-terminus at the 3' hydrophobic cleft (also see Supplementary Fig. 4a). Hydrogen-bonding and van der Waals interactions are shown as dashed blue and dashed yellow lines, respectively. **d**, **e** Comparison of the 3' hydrophobic cleft of the 3' dyskerin (**d**) with the analogous region at the 5' dyskerin subunit (**e**).

binding is also stabilized by electrostatic interactions between residues D26 of helix 1 of the 5' dyskerin and K43 of the 3' dyskerin (Fig. 2d–f). While the most N-terminal parts of dyskerin (residues 1-22 and 1-42 for the 5' and 3' dyskerin, respectively) are not visible in our structure, several disease mutations are located at our newly resolved inter-dyskerin interface (Fig. 2g)[16]. Mutations that perturb the hydrophobic cleft (e.g. F36V) or introduce charge-reversals (e.g. K43E) would likely disrupt the hydrophobic and electrostatic inter-dyskerin interactions and hence telomerase RNP assembly (Fig. 2f, g, and Supplementary Fig. 6a)[17,38–43]. Therefore, our structure rationalizes the structural and functional effects of these disease mutations.

Helix 1 of the 5' dyskerin NTE also interacts with the 3' NOP10, further enhancing the inter-heterotetramer interactions (Fig. 3a–c, and Supplementary Fig. 4a). The C-terminus of the 3' NOP10 interacts with the hydrophobic cleft of the 3' dyskerin (as described above, Fig. 2c, f) and stabilizes helix 1 of the 5' dyskerin (Fig. 3b–d). The C-terminal carboxyl group of the 3' NOP10 hydrogen bonds with both conserved Q31 of the 5' dyskerin helix 1 and the highly conserved K57 of the 3' dyskerin NTE (Fig. 3c and Supplementary Fig. 6a). Mutations at Q31 (Q31E, Q31K) are associated with dyskeratosis congenita and would not support the described interactions with the 3' NOP10[44,45]. This underscores the importance of interactions at this region for telomerase H/ACA RNP stability[44,45].

To further validate the new inter-dyskerin interactions observed, we supplemented our telomerase reconstitution with overexpressed 3xFLAG-tagged wild-type or disease mutant dyskerin (I30M, Q31E, Q31K, F36V, K43E and L56S) (Fig. 2g) and performed oligonucleotide affinity purification via hTR (Methods and Supplementary Fig. 7). Under normal reconstitution conditions, telomerase assembles with endogenous dyskerin[46]. By comparing the relative intensities of the overexpressed and the endogenous dyskerin signals in the immunoblots, we observe that overexpressed wild-type 3xFLAG-tagged

dyskerin efficiently outcompetes endogenous dyskerin for incorporation into telomerase (Supplementary Fig. 7b–d). In contrast, all the dyskerin mutants compete less efficiently with the endogenous dyskerin for incorporation into telomerase despite being overexpressed (Supplementary Fig. 7b–d). This is likely caused by the weakened inter-dyskerin interactions, agreeing with our structural observations.

Interestingly, the NTE and TruB domains of the 5' dyskerin do not form a hydrophobic cleft like their 3' counterparts (Fig. 3e). We did not observe analogous interactions between the 3' dyskerin and the C-terminus of the 5' NOP10 (Fig. 3d, e). The same region that forms helix 1 in the 5' dyskerin subunit (Fig. 3d) is disordered in the 3' dyskerin subunit in our structure and does not bind at the 5' dyskerin-NOP10 interface (Fig. 3e).

Unlike the canonical snoRNPs and scaRNPs, the two hairpins of telomerase H/ACA RNA domain have different affinities for the H/ACA heterotetramers (Fig. 1a, c)[9,12]. The 3' RNA hairpin interacts with the dyskerin, NHP2 and NOP10 subunits of the 3' heterotetramer, as previously seen in single-hairpin H/ACA RNP structures (Fig. 1a)[27,28]. On the other hand, the 5' H/ACA RNA hairpin is bound only by the 5' dyskerin of the 5' heterotetramer (Fig. 1a)[27,28]. Our double-hairpin telomerase H/ACA RNP structure provides the most extensive description of the inter-heterotetramer interactions to-date. The association of the 5' heterotetramer requires the inter-heterotetramer interactions to compensate for its sub-optimal binding to the 5' hairpin telomerase H/ACA RNA[9,47], explaining why the enhancement for inter-dyskerin interactions by NOP10 is only required for the 5' dyskerin subunit (Fig. 3a)[10]. Therefore, the disease mutations that disrupt the observed protein-protein interactions would compromise the binding of the 5' heterotetramer on telomerase 5' H/ACA hairpin, resulting in telomerase deficiency and telomere maintenance defects (Fig. 2g)[38,39,41–43,44,45,48].

### The dyskerin-hTR interface rationalizes disease mutations

Our structure allows modeling of additional residues of hTR, revealing novel interactions between dyskerin and hTR. As seen in previous structures[10,35], the conserved H and ACA boxes of hTR are brought into proximity by the two dyskerin subunits (Fig. 4a and Supplementary Fig. 3a, b). Here we found that the H box nucleotides are buried within the 5′ dyskerin (Fig. 4b) and contacted by a region termed the H box binding motif (HBM) (residues 390-421 of the 5′ dyskerin) (Fig. 4c, Supplementary Fig. 4f, and Supplementary Data 1). Residues of the HBM are associated with telomere-related diseases (Fig. 4d)[38,41,49–55]. Our structure suggests that these disease mutations likely affect telomerase function by destabilizing the overall fold of the dyskerin HBM, and thus interactions with the H box.

In yeast, the HBM of Cbf5 (dyskerin orthologue) interacts with SHQ1 (PDB 3UAI [https://doi.org/10.2210/pdb3UAI/pdb]), which is an assembly factor involved in early H/ACA biogenesis in both yeast and human (Supplementary Fig. 8a, b)[56,57]. AlphaFold2 predicts positioning of the human HBM domain on the SHQ1 in a similar manner to that observed in yeast (Supplementary Fig. 8c–e)[58]. Mutations in SHQ1 residues (e.g. A426) that impair interaction with dyskerin (Supplementary Fig. 8d) have been described in patients with severe neurological disorders, resembling the HH syndrome[59]. Thus, mutations in the HBM would also affect telomerase biogenesis by disrupting SHQ1 binding (Fig. 4d and Supplementary Fig. 8).

The ACA box is positioned within a cavity formed by the 3′ dyskerin (Fig. 4e). Here we resolved an additional nucleotide following the ACA box at the 3′ end of hTR (G450) (Fig. 4e and Supplementary Fig. 4e)[10,35]. In our structure, G450 base stacks with conserved residue H68 of the 3′ dyskerin and interacts with residue S42 of the 5′ dyskerin NTE (Fig. 4f and Supplementary Fig. 4e). Mutations at G450 cause telomere length defects in vivo[60,61]. Mutations at H68 (H68R, H68Q and H68Y) and its surrounding residues (T66 and T67) lead to dyskeratosis congenita and HH syndromes (Fig. 4g)[38,41,49–51,62–64]. These functional data suggest that disrupting the interaction between H68 and G450 affects telomerase function in vivo.

To further validate the dyskerin-hTR interaction described above, we again supplemented our reconstituted telomerase with overexpressed 3xFLAG-tagged dyskerin with H68A or T66A/T67A/H68A mutations (Supplementary Fig. 7a). Upon oligonucleotide-affinity purification via hTR, both H68A and T66A/T67A/H68A mutants are incorporated into telomerase less efficiently compared to the overexpressed wild-type counterpart, evident in the lower relative intensities compared to endogenous dyskerin in the immunoblots (Supplementary Fig. 7b–d). Additionally, we reconstituted telomerase with G450 base substitutions and performed in vitro telomerase activity assays using these mutants (Supplementary Fig. 9). For these experiments, to avoid bypassing 3′ end processing of hTR, we also used a different expression plasmid of hTR, in which the hepatitis delta virus ribozyme sequence is replaced with 500 nucleotides of the endogenous 3′ untranslated region (UTR) of hTR[46] (Methods). The mutations resulted in slight yet significant reductions in telomerase activity compared to wild-type telomerase (Supplementary Fig. 9c). The observed effects are small because all alternative nucleotides can maintain the stacking interaction with H68 of the 3′ dyskerin to some extent while disrupting the hydrogen bond with residue S42 of the 5′ dyskerin (Fig. 4f). The activity reductions were more pronounced in G450C and G450U than in G450A because a purine base substitution stacks with H68 more efficiently than a pyrimidine substitution (C or U). Overall, our mutagenesis experiments are in good agreement with the structural observations and previous functional data[38,41,49–51,60–64].

### A more complete model of TCAB1 defines interactions with hTR and NHP2

TCAB1 is essential for not only hTR accumulation but also telomerase assembly because TERT and hTR localize to different nuclear compartments in the absence of TCAB1[65,66]. Our structure reveals the most complete model for TCAB1 so far and new insights into previously described interactions that TCAB1 makes with hTR and the 3′ NHP2 (Fig. 5b)[10,34,35].

We resolved a β-hairpin loop of TCAB1 (residues 317-326), which contacts the major groove backbone of the P8 stem-loop of hTR via residues K321 and S325 (Fig. 5a–c, and Supplementary Fig. 4g), stabilizing the P8 stem-loop conformation. TCAB1 also binds hTR at the CAB box (a conserved ugAG sequence), located within the P8b stem-loop (Fig. 5a, b, d). We observe base-specific interactions between TCAB1 and conserved nucleotides A413 and G414 of the CAB box, which would be poorly supported by any other bases (Fig. 5d)[12,15,66,67]. This explains why substitutions of CAB box nucleotides lead to disruption of hTR localization and telomerase activity in vivo[66–68].

We also identified direct interactions between TCAB1 and the 3′ NHP2 (Fig. 5b, e). TCAB1 residues 483-489, herein termed the NHP2-interacting loop (NIL), form extensive interactions with the 3′ NHP2 subunit (Fig. 5e and Supplementary Data 1). Residue E487 of TCAB1 forms a salt-bridge with K52 of the 3′ NHP2, whilst P488 of TCAB1 interacts hydrophobically with A116 and A117 of the 3′ NHP2. These interactions could play a role in enhancing TCAB1 association with the H/ACA RNP.

### Consensus map reveals a non-catalytic semi-closed conformation of the 3′ dyskerin thumb loop

The archaeal dyskerin thumb loop adopts a closed conformation in substrate-bound H/ACA RNP structures[27,28]. In this conformation, the thumb loop interacts with the substrate RNA and locks the target U nucleotide into the pseudouridylation active site of dyskerin containing the catalytic aspartate residue (D85 in archaea, D125 in human)[69]. In the absence of a substrate and during product release, the archaeal thumb loop switches to an open conformation, rotating away from the active site to interact extensively with Gar1 (GAR1 in humans)[30,31,70]. Furthermore, the thumb loop and its interactions have not been fully resolved in any previous eukaryotic H/ACA RNP structures, including telomerase[10,32,34,35].

Our telomerase holoenzyme sample was prepared without the addition of an RNA substrate for pseudouridylation. Therefore, we had anticipated to find the dyskerin thumb loop in an open conformation. However, focused classification revealed a subset of particles with a well-resolved thumb loop for the 3′ dyskerin (see Methods, Supplementary Fig. 1 and Supplementary Fig. 3e, inset). Surprisingly, we observed that the 3′ dyskerin thumb loop adopts a nearly closed, or "semi-closed", conformation (Fig. 6a and Supplementary Data 1), resembling the closed conformation observed in the substrate-bound archaeal complex (Fig. 6d, e)[27,28]. However, this semi-closed state appears to be an intermediate state prior to closure of the substrate into the active site for catalysis. We observe direct interactions between the 3′ dyskerin thumb loop and an unpaired region of hTR (Fig. 6b and Supplementary Fig. 4d). Nucleotide G393 of hTR flips out of the RNA duplex towards the dyskerin active site (Fig. 6b), appearing analogous to the target U nucleotide in a canonical H/ACA RNP (Fig. 6d, e). Compared to the archaeal structures, the thumb loop and hTR are further away from the pseudouridylation active site (Fig. 6d, e). Thus, this conformation likely represents an intermediate substrate-loading state, with nucleotide G393 in a position incompatible with catalysis (Fig. 6d, e).

The yeast Cbf5 (dyskerin) thumb loop and its interactions with the first helix (helix 1) of the Gar1 (GAR1 in human) C-terminal extension (CTE) were shown to play a critical role in substrate turnover during pseudouridylation[32]. In archaea, Gar1 is thought to sense the state of the pseudouridylation catalytic cycle and control the conformation of the dyskerin thumb loop accordingly[29]. In our structure, the base of the thumb loop of the 3′ dyskerin interacts with a hydrophobic surface on

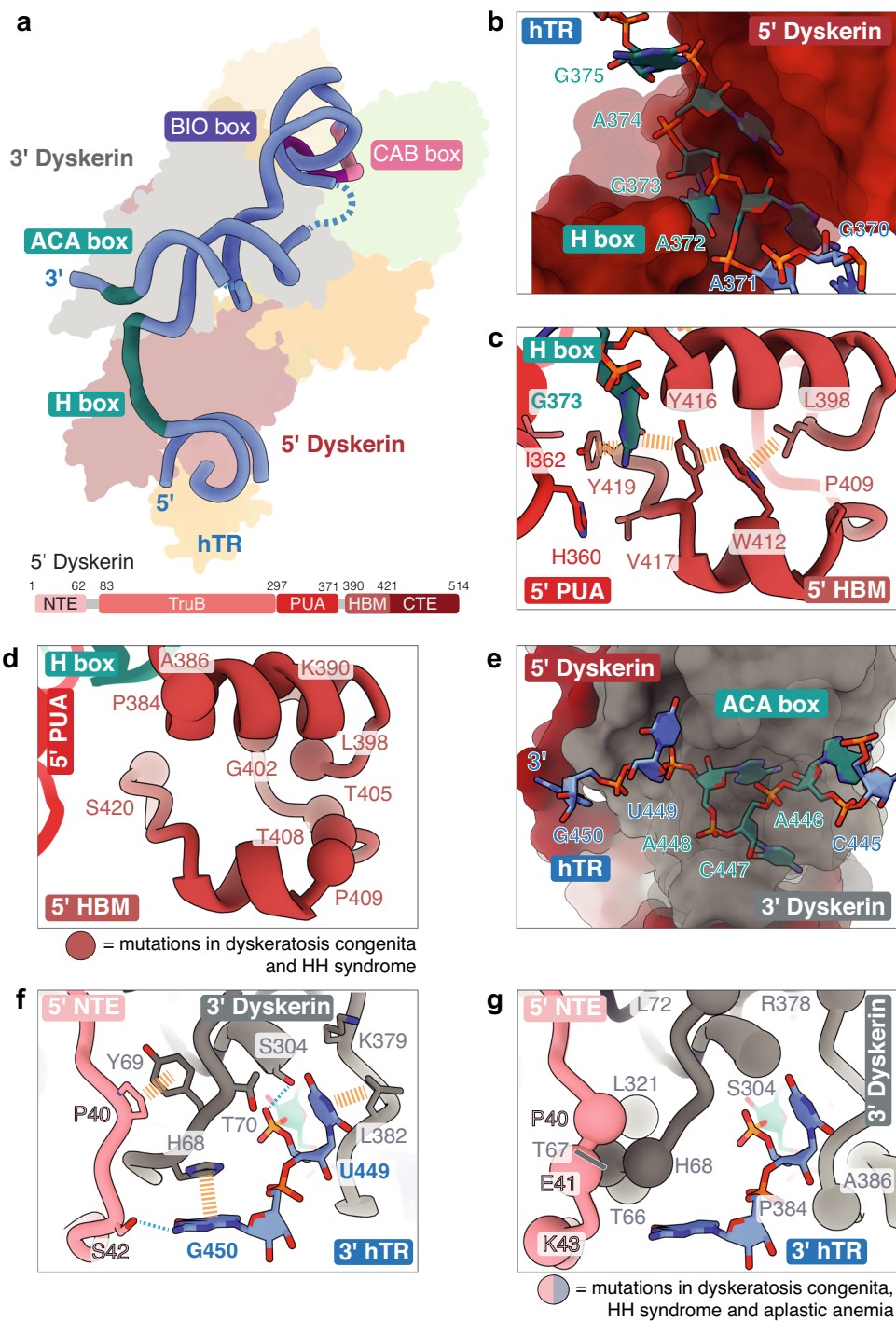

**Fig. 4 | Dyskerin-hTR interface rationalizes disease mutations. a** Structure of the H/ACA domain of hTR. The structure is overlayed atop the H/ACA proteins. Conserved H, ACA, CAB, and BIO motifs are indicated. Unresolved nucleotides are shown as blue dashed lines. **b** Positioning of the H box of hTR in a cavity formed by the 5′ dyskerin. **c** Interactions between residue G373 of the H box and the H box binding motif (HBM) of the 5′ dyskerin as part of the CTE domain (also see Supplementary Fig. 4f). Stacking interactions are shown as dashed yellow lines. Domains of the 5′ dyskerin are colored as shown in the schematic. **d** Dyskeratosis congenita and HH syndrome disease mutations in the HBM of the 5′ dyskerin.

Disease-associated residues are highlighted as spheres. **e** Positioning of the ACA box of hTR in a cavity formed by the 3′ dyskerin. The 3′ end of hTR (G450) reaches towards the 5′ dyskerin. **f** Interactions between the 5′ and 3′ dyskerin, and the 3′ end of hTR (also see Supplementary Fig. 4e). Hydrogen-bonding and stacking interactions are shown as dashed blue and dashed yellow lines, respectively. **g** Dyskeratosis congenita, HH syndrome, aplastic anemia, and bone marrow failure disease mutations in the region shown in (**f**). Disease-associated residues are highlighted as spheres.

the 3′ GAR1 (Fig. 6c and Supplementary Fig. 4c). Helix 1 of the 3′ GAR1 CTE positions below the thumb loop base to extend the hydrophobic surface (Fig. 6c and Supplementary Data 1). The position of GAR1 helix 1 and its interactions with the 3′ dyskerin in humans are analogous to those found in the yeast H/ACA RNP structure (Supplementary Fig. 6b and Supplementary Fig. 10a, b)[32]. Although the rest of the 3′ GAR1 CTE remains unresolved, our structure suggests that GAR1 helix 1 likely serves a similar role in substrate turnover in human H/ACA RNPs[32].

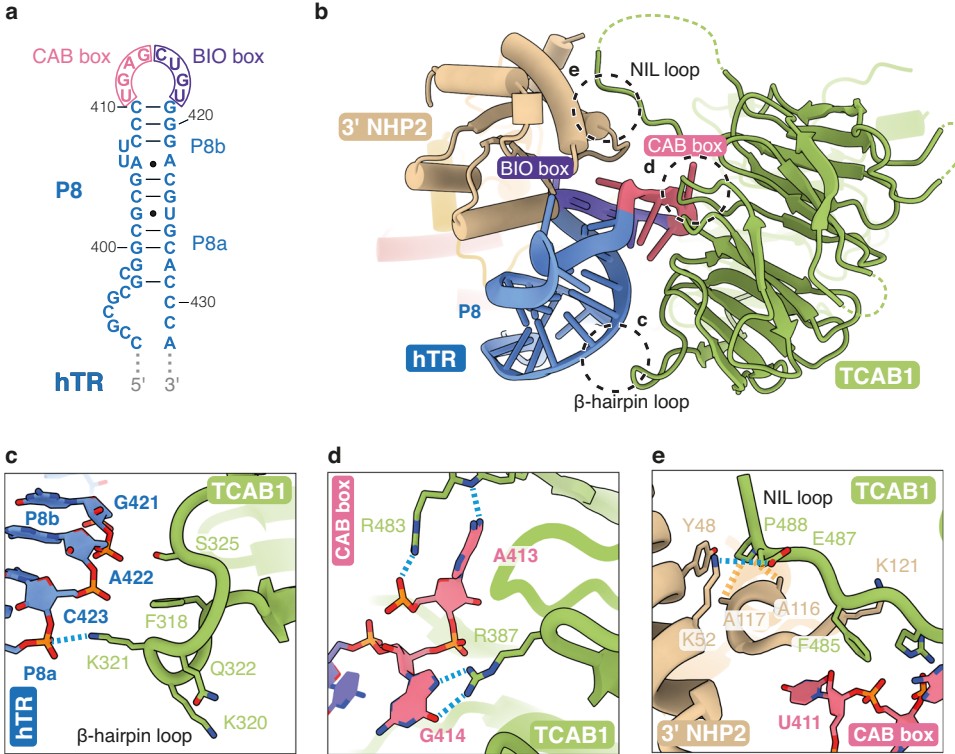

**Fig. 5 | TCAB1 interaction with hTR and the 3' NHP2. a** Secondary structure of the P8 stem-loop of hTR, with CAB and BIO box labeled. Watson-Crick-Franklin base pairs are indicated as solid lines. Non-Watson-Crick-Franklin base pairs are indicated as dots. **b** Interactions of TCAB1 with the BIO and CAB boxes of hTR and the 3' NHP2. Circles indicate the three contact regions depicted in (**c**–**e**). Unresolved loops are shown as green dashed lines. **c** Close-up view of the β-hairpin loop of TCAB1, and its interactions with the P8 stem of hTR (also see Supplementary Fig. 4g). **d** Close-up view of the TCAB1-CAB box interaction. Certain nucleotides of the CAB and BIO boxes are transparent for clarity. **e** Close-up view of the interactions between the NHP2 interacting loop (NIL) of TCAB1 and the 3' NHP2. Hydrogen-bonding and van der Waals interactions are shown as dashed blue and dashed yellow lines, respectively.

Despite using a similar classification approach, the thumb loop of the 5' dyskerin remained poorly resolved. The region where the thumb loop of the 5' dyskerin would be faces the solvent in our structure. Interestingly, the CTE helix of the 5' GAR1 is in a similar position relative to the putative 5' dyskerin active site compared to the 3' counterpart (Supplementary Fig. 10d). It is possible that the weaker interactions with hTR result in the increased mobility of the 5' dyskerin thumb loop.

### hTR mimics the substrate-guide RNA duplex of a H/ACA RNP
During pseudouridylation, the guide sequences of the H/ACA RNA pseudouridylation pocket base-pair with the target substrate RNA (Fig. 7a). Despite the differences in substrate and guide RNA sequences, the resulting RNA architecture is similar across all H/ACA RNA and RNP structures characterized to-date[25–28]. The bound substrate adopts a U-like conformation by forming two RNA duplexes with the 5' and 3' pseudouridylation guide sequences (PS1 and PS2 helix, respectively in Duan et al.[27]) (Fig. 7a, b). The "UN" dinucleotide of the substrate is unpaired and located at the apex of the U-like conformation. RNA-protein interactions normally position this dinucleotide into the pseudouridylation active site of dyskerin[70].

Interestingly, despite the lack of any substrate RNA in the telomerase H/ACA RNP, the 3' hairpin pseudouridylation pocket of our structure exhibits partial similarities to that of the canonical substrate-bound H/ACA RNP. In the published substrate-bound archaeal H/ACA RNP structure[27,28], the PS2 RNA helix is formed by the substrate and the 3' guide sequence of the pseudouridylation pocket (Fig. 7a, b). Whereas, in our structure, part of the P7 stem of the hTR 3' hairpin mimics the PS2 helix (Fig. 7c, d, and Supplementary Data 1). hTR, however, lacks an analogue to the PS1 helix. Nucleotides 394-398 of hTR are disordered and likely too short to form a helix resembling PS1

(Fig. 7c, d). On the other hand, the pseudouridylation pocket of the telomerase RNA H/ACA 5' hairpin is highly flexible and does not appear to mimic a PS2 helix near the 5' dyskerin like its 3' counterpart. Together, our structure reveals that hTR acts as a pseudosubstrate at the 3' dyskerin. Thus, we call the P7 part of hTR that mimics the substrate strand of the PS2 helix the "pseudosubstrate" region (Supplementary Data 1). Although previous work also suggested that hTR can act as a pseudosubstrate[35], the 3' guide sequence was previously misassigned as the "pseudosubstrate" strand (Fig. 7d). The ability of hTR to mimic a substrate-bound RNA explains why we observe the 3' dyskerin thumb loop in a semi-closed conformation rather than an open conformation.

### Heterogeneity analysis reveals an open conformation of the 3' dyskerin thumb loop
In archaeal complexes, a change of the Cbf5 (dyskerin) thumb loop from an open to a closed conformation is critical to the pseudouridylation catalytic cycle[27,28]. However, conformational changes of the eukaryotic dyskerin thumb loop have not been structurally characterized. During early stages of image processing, we noticed the lower local resolution estimates near the 3' dyskerin thumb loop. Therefore, we asked if other thumb loop conformations were present in our data. To explore this, we performed 3D variability analysis (3DVA) in cryoSPARC (Supplementary Fig. 11)[71].

Our data reveal the dynamic nature of the dyskerin thumb loop in vertebrate telomerase H/ACA RNP, for the first time. Globally, 3DVA shows a rotation of the two H/ACA heterotetramers relative to one another (Fig. 8a). Locally, 3DVA combined with further classification and refinement shows two states (State 1 and State 2) of the 3' dyskerin thumb loop and hTR (Fig. 8a, b, and Supplementary Figs. 11, 12a).

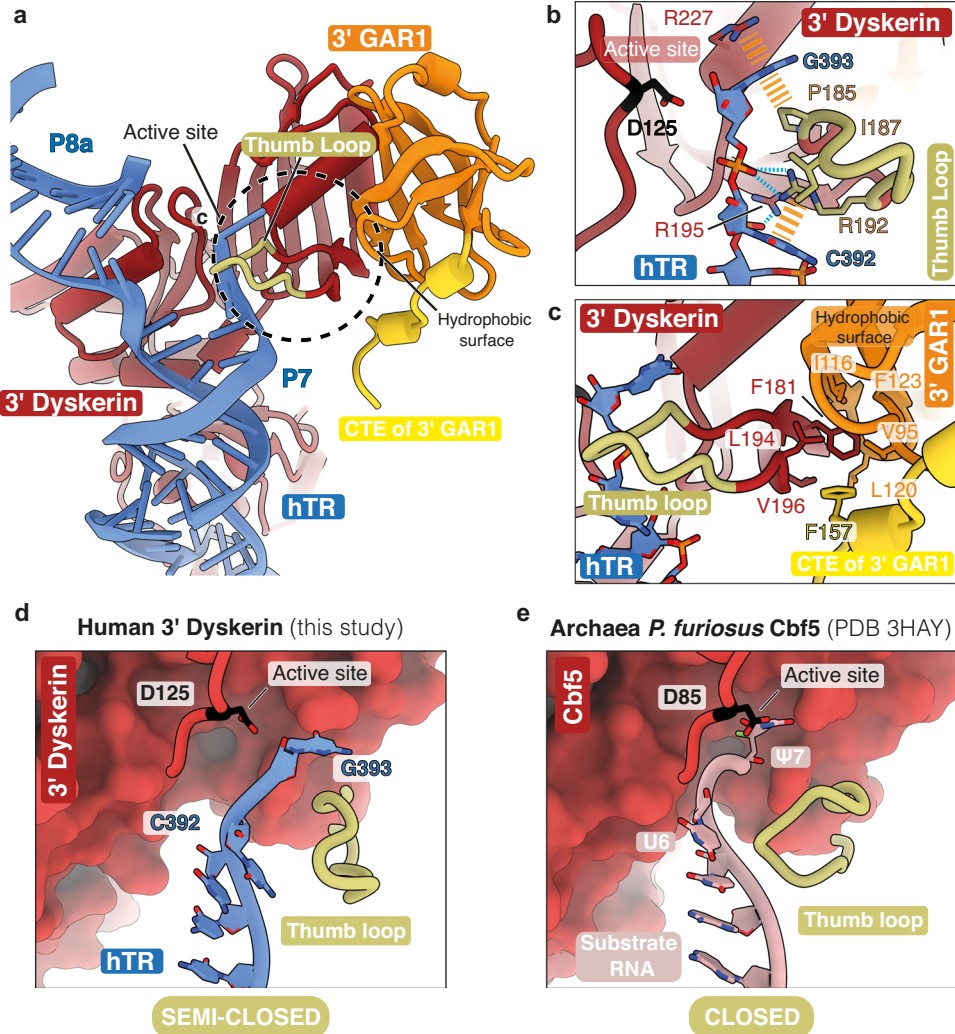

**Fig. 6 | Interactions at the 3' dyskerin thumb loop. a** Interactions between the 3' dyskerin thumb loop, hTR and the 3' GAR1. The circle indicates the 3' dyskerin thumb loop interaction depicted in (**b**, **c**). The eukaryote specific CTE of GAR1 is colored yellow (Supplementary Fig. 6b). **b** Close-up view of the interaction between the 3' dyskerin thumb loop and hTR (also see Supplementary Fig. 4d). The side-chain of the catalytic aspartate (D125) is shown in the dyskerin active site. Hydrogen-bonding and van der Waals interactions are shown as dashed blue and dashed yellow lines, respectively. **c** Close-up view of the interactions between the base of the 3' dyskerin thumb loop and the 3' GAR1 (also see Supplementary Fig. 4c). **d**, **e** Comparison of the human 3' dyskerin active site and thumb loop with the substrate-bound archaeal Cbf5 from *Pyrococcus furiosus* (*P. furiosus*) (PDB 3HAY [https://doi.org/10.2210/pdb3HAY/pdb])[27], respectively. The active site and the catalytic aspartate residue are indicated. In the Cbf5 structure shown in (**e**), the pseudouridylated residue (ψ7) of the substrate RNA is positioned in the active site. In our structure shown in (**d**), the dyskerin active site is empty.

State 1 resembles the consensus refinement with the thumb loop in the semi-closed conformation as described above (Fig. 8a, State 1, c). In State 2, the thumb loop is rotated away from hTR, towards the 3' GAR1 subunit (Fig. 8a, State 2, d, and Supplementary Data 2). This is accompanied by an outward movement of the hTR P7 duplex and a rotameric switch of F98 of the 3' GAR1 hydrophobic surface (Fig. 8d, e). hTR moves out of the 3' dyskerin active site. Therefore, we propose that State 2 represents the eukaryotic analog of the open state described for archaeal H/ACA RNP complexes (Fig. 8a, State 2, d, and Supplementary Fig. 10c)[31].

Interestingly, the thumb loop in our open state adopts a significantly less open conformation compared to the archaeal counterpart (Supplementary Fig. 10c, b). To understand this difference, we compared our open state structure with the reported yeast Cbf5-Nop10-Gar1 ternary complex (Supplementary Fig. 10)[32]. In both structures, we observe that the eukaryote-specific helix 1 of GAR1 blocks the thumb loop from adopting an archaeal-like open conformation (Supplementary Fig. 10). Consequently, the conformational switch between the eukaryotic open (substrate-free or product-release) and closed (substrate-bound) states of the dyskerin thumb loop appears more subtle than that of the archaeal H/ACA RNPs[27]. Together, our data provides insights into the dynamics of the thumb loop, which has important implications for its role in coordinating the dyskerin active site during eukaryotic H/ACA pseudouridylation.

## Discussion

Since its discovery nearly four decades ago[72,73], telomerase has been viewed as an important drug target for anti-cancer and aging therapeutics[74–81]. However, rational drug design against telomerase has been slowed by the lack of high-resolution structural information. The flexibility, low abundance, and complexity of human telomerase present a challenge to high-resolution structural determination. The work presented here uses recent technological advances in cryo-EM to determine a 2.7 Å resolution structure of the human telomerase H/ACA RNP lobe and address the inherent conformational flexibility within this part of telomerase. The resolution achieved is reaching the range

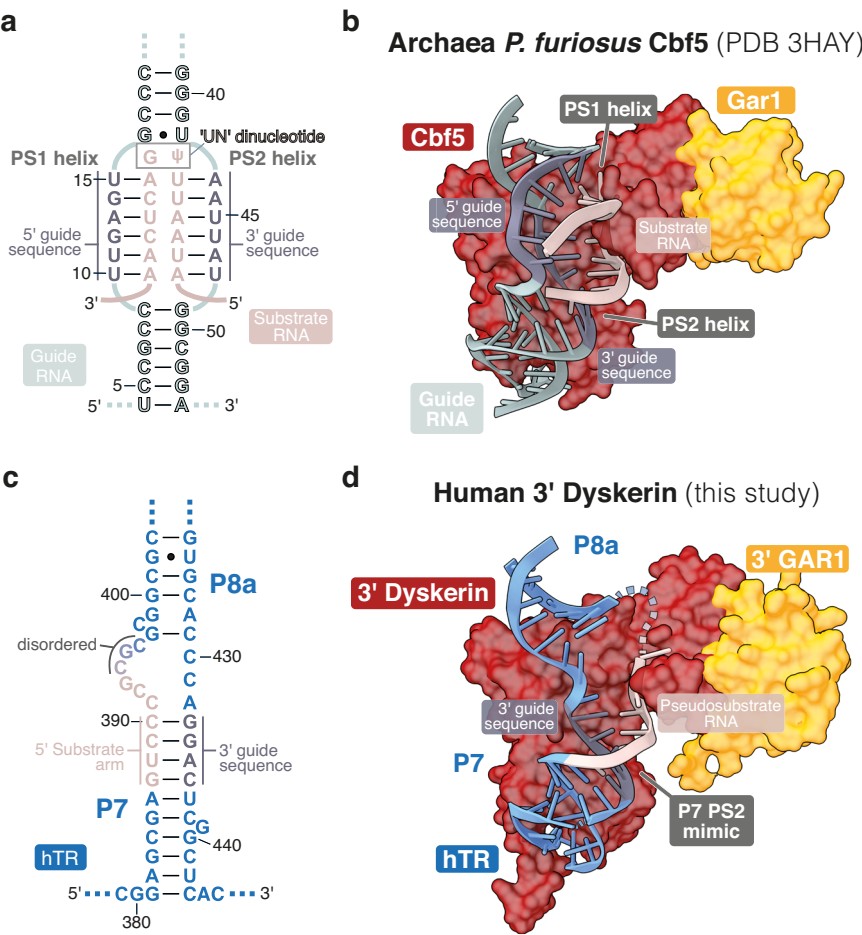

**Fig. 7 | Mimicry of substrate RNA binding by hTR. a** Secondary structure of the guide RNA (grey) in complex with the substrate RNA (pink) in an archaeal H/ACA RNP (PDB 3HAY [https://doi.org/10.2210/pdb3HAY/pdb]) shown in (**b**)[27]. The PS1 and PS2 helices are shown with the 5′ and 3′ guide sequences respectively. The unpaired 'UN' dinucleotide is labeled. Watson-Crick-Franklin base pairs are indicated as solid lines. Non-Watson-Crick-Franklin base pairs are indicated as dots. **b** Structure of the guide (grey) and substrate (pink) RNAs in an archaeal H/ACA RNP (PDB 3HAY [https://doi.org/10.2210/pdb3HAY/pdb])[27]. The 5′ and 3′ guide sequences are colored in dark-grey. The substrate-guide duplexes (PS1, PS2) are indicated. **c** Secondary structure of the 3′ hairpin of the hTR H/ACA domain in our structure shown in (**d**). Parts of hTR that resemble the 3′ guide sequence and the substrate RNA are colored in dark-grey and in pink, respectively. **d** Structure of hTR on the 3′ dyskerin observed in our structure. Regions of hTR are colored as described in (**c**).

amenable to structure-based drug design, holding great future promise for telomerase-targeted therapeutics.

We note that the H/ACA RNP of telomerase, especially the inter-dyskerin interface, is a hotspot for disease-associated mutations[9,10]. The new molecular interactions described here allow for a more complete understanding of the pathogenesis of telomere-related disease mutations (Figs. 2–4). Such knowledge is critical for future studies involving the manipulation of interactions within the H/ACA RNP for disease treatment. Using small molecules to stabilize protein-protein interactions is an emerging strategy in drug design and may be applied for telomerase[82]. For example, molecules that stabilize the inter-dyskerin interface may be used to treat telomerase deficiency in patients with dyskeratosis congenita or HH syndrome. On the other hand, the telomerase 5′ H/ACA heterotetramer has an increased dependence on the inter-dyskerin and inter-heterotetramer interfaces compared to canonical H/ACA RNPs. Thus, therapeutics designed to disrupt these interactions could specifically inhibit telomerase biogenesis in cancer cells, with minimal effects to other cellular H/ACA RNPs.

Our data shows that hTR acts as a pseudosubstrate, appearing to stall the 3′ dyskerin thumb loop in a substrate-loading (semi-closed) conformation (Fig. 7). It remains unknown if telomerase possesses pseudouridylation activity in cells[83]. Furthermore, it is possible that the substrate mimicry by the telomerase RNA serves to inhibit telomerase from unwanted pseudouridylation of cellular RNAs. This explains why artificial substrates that match telomerase hTR pseudosubstrate sequence are not pseudouridylated while the substrates that match other H/ACA RNPs are[84,85].

Despite the lack of a known pseudouridylation substrate for telomerase, we took advantage of modern cryo-EM data processing techniques to resolve two conformations of the 3′ dyskerin thumb loop present within our telomerase dataset. Our telomerase H/ACA RNP structures provide a model for exploring the conformational plasticity of the eukaryotic dyskerin thumb loop. The two states resemble the substrate loading and the product-release state of archaeal H/ACA RNP (Fig. 8f–g). Our structures illuminate features that are specific to the eukaryotic H/ACA pseudouridylation system, namely: the interactions between the eukaryotic specific helix 1 of GAR1 and the thumb loop (Fig. 6c); and the subtle differences between the open and closed conformation of the thumb loop compared to archaeal H/ACA RNPs (Fig. 8c, d). In archaea, the large conformational switch of the thumb loop from the closed to the open state is thought to promote substrate turnover by peeling away the product RNA (Supplementary Fig. 10c)[86]. Given the small differences between the observed thumb loop conformations (Supplementary Fig. 10b), it will be interesting to see how substrate turnover is achieved in canonical eukaryotic H/ACA systems.

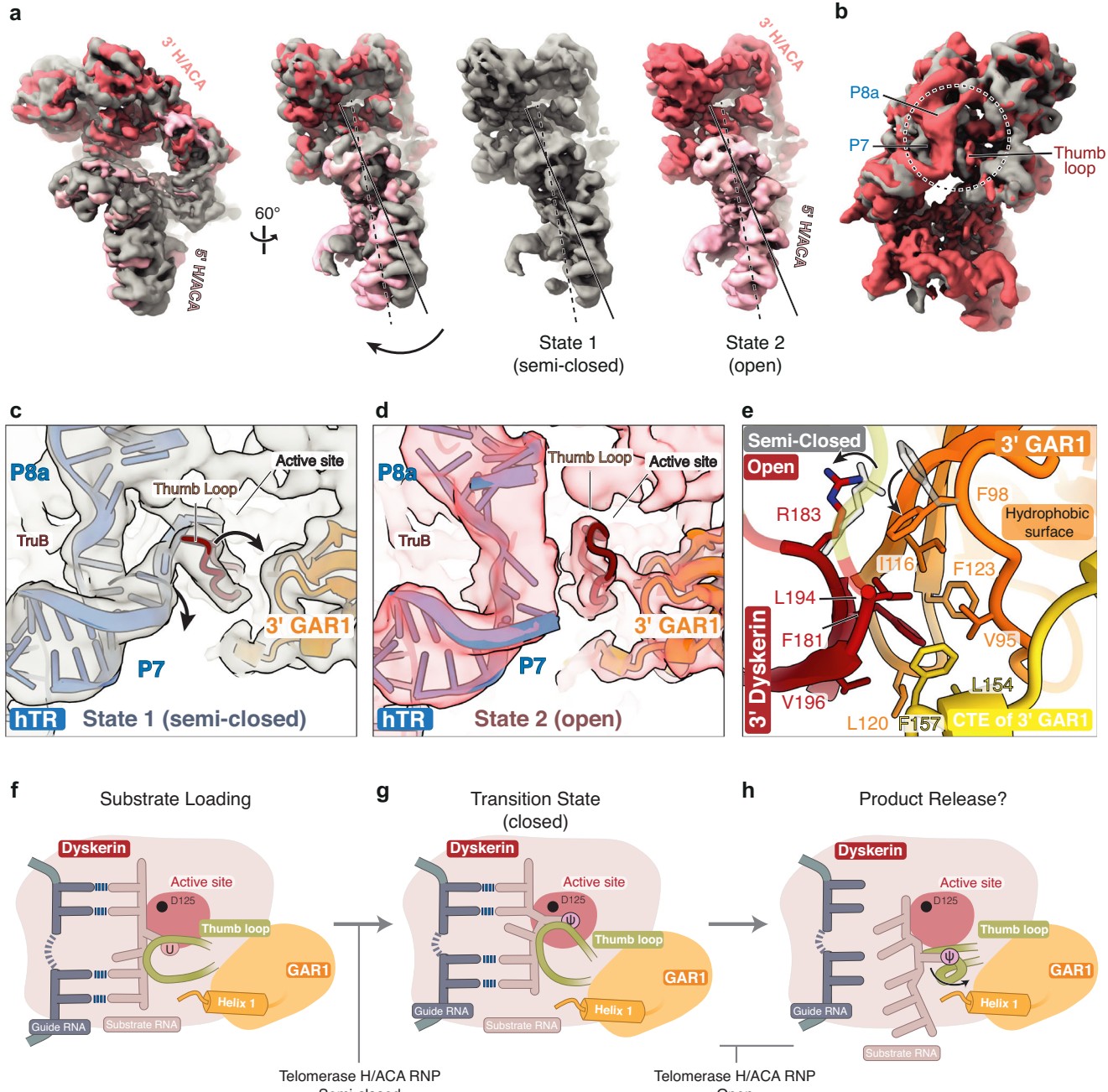

**Fig. 8 | 3D variability analysis (3DVA) reveals conformational heterogeneity at the 3′ dyskerin thumb loop. a** Comparison of the 3DVA conformations (State 1 and State 2). The arrow indicates the degree of rotation/movement in State 2 relative to State 1. **b** An overlay of State 1 and State 2. The circle indicates the variability around the 3′ dyskerin thumb loop, including the P7 and P8a stems of hTR. **c** Close-up view of the 3′ dyskerin thumb loop in a semi-closed conformation. Arrows indicate the movement of hTR and the thumb loop from the semi-closed conformation to the open conformation described in (**d**). **d** Close-up view of the dyskerin thumb loop resembling the open conformation. Atomic models shown in (**c**) and (**d**) come from high resolution reconstructions (Supplementary Fig. 2, 11) and are fitted into the low resolution 3DVA densities. **e** An overlay of the thumb loop interaction with the 3′ GAR1 in the open conformation (colored) and the semi-closed conformation (grey) (also see Supplementary Fig. 12f). Residues of the 3′ GAR1 hydrophobic surface are shown. Changes from the semi-closed to open conformation are indicated with an arrow. Certain residues of the 3′ dyskerin thumb loop are transparent for clarity. **f–h** Schematics of proposed pseudouridylation cycle in eukaryotic H/ACA RNPs. The cycle includes substrate loading (**f**), pseudouridylation at the transition state (**g**) and product release (**h**). The dyskerin thumb loop changes conformation from partially closed (**f**) to fully closed (**g**) to open (**h**). The extent of thumb loop opening during product release is unknown and hence indicated by a question mark (?). Proposed states observed for the telomerase H/ACA RNP are indicated.

In substrate-free archaeal H/ACA RNPs, the guide RNA sequences are largely disordered[31]. However, upon substrate binding, the PS2 helix makes nearly all of the newly formed interactions with Cbf5 (dyskerin orthologue), unlike the PS1 helix (Fig. 7b)[33]. Interestingly, we observe that hTR mimics the high-affinity PS2 helix at the 3′ hairpin of hTR, rather than the low affinity PS1 helix.

Previous work has shown that the 3′ hairpin is essential for telomerase H/ACA RNP assembly, whereas the 5′ hairpin is evolutionarily divergent and more tolerant to changes[47,87]. Therefore, we hypothesize that mimicry of the PS2 helix within the 3′ hairpin ensures a more stable association with dyskerin, further enhancing telomerase assembly.

## Methods

### Telomerase expression and purification

Human telomerase reconstitution was carried out in Sekne et al. [36], as previously described[9]. Briefly, telomerase was reconstituted in HEK293T cells (ATCC, Cat# CRL-3216, RRID:CVCL_0063) using pcDNA3.1-ZZ-TEV-Twin-Strep-TERT and pcDNA 3.1-U3-hTR-HDV, where HDV is the hepatitis delta virus ribozyme, and TEV is the Tobacco Etch Virus protease cleavage site[9,36]. Whole-cell extracts from 120-160 15 cm plates of cells were prepared by three hypotonic freeze-thaw cycles, snap-frozen in liquid nitrogen and stored at −80 °C until use. Telomerase lysates were incubated with streptavidin agarose resin (Merck, Cat# E5529) prebound to a 5' biotinylated 2'-O-methyl oligonucleotide[88] (sequence: CTAGACCTGTCATCAmGmUmUmAm GmGmGmUmUmAmG where m stands for 2'-O-methyl modification) for 3 hours at room temperature. After washing the resin with wash buffer (20 mM HEPES NaOH pH 8.0, 150 mM NaCl, 2 mM MgCl$_2$, 0.2 mM EGTA, 10% glycerol, 0.1% IGEPAL CA-630, 0.2 mM PMSF and 1 mM DTT), the sample was eluted with a competitor oligonucleotide[88] (sequence: CTAACCCTAACTGATGACAGGTCTAGddC where ddC stands for dideoxycytosine). The eluate was added to the pre-washed MagStrepXT resin (IBA LifeSciences, Cat# 2-4090-002) and incubated overnight at 4 °C. After washing the resin with wash buffer, complexes were eluted by incubation in biotin elution buffer (100 mM HEPES NaOH pH 8.0, 150 mM NaCl, 1 mM EDTA, 10 mM biotin, 2 mM MgCl$_2$, 10% glycerol, 0.1% IGEPAL CA-630, 0.2 mM PMSF and 1 mM DTT) for 1 hour at 4 °C.

### Purification of TPP1-POT1-TIN2 (TPT)

TPP1-POT1-TIN2 (TPT) purification was carried out previously, as described in Sekne et al. [36]. Briefly, TPT was expressed in *Spodoptera frugiperda* (Sf9) (Sf9, Oxford Expression Technologies Ltd, Cat# 600100) cells and purified as previously described[36]. Briefly, baculoviruses expressing TPP1, POT1, and TIN2 were used to infect 1 L of Sf9 cells, which were grown for 72 h at 27 °C before being harvested by centrifugation. Cell pellets were resuspended in lysis buffer (25 mM HEPES NaOH pH 8.0, 300 mM NaCl, 1 mM MgCl$_2$, 0.01 mM ZnSO$_4$, 0.01% IGEPAL CA-630, 1 mM PMSF, 1 mM DTT and 1x cOmplete protease inhibitor tablets (Roche, Cat# 11873580001)). Lysates were sonicated, ultracentrifuged, filtered and bound to dextrin Sepharose resin (Cytiva, Cat# 28-9355-97), followed by an overnight elution by on-column cleavage with SUMOstar protease (LifeSensors, Cat# SP4110). The sample was further purified on a 5 mL HiTrap heparin HP column (Cytiva, Cat# 17040701) and a HiLoad 16/600 Superdex 200 column (Cytiva, Cat# 28-9893-35) in SEC buffer (25 mM HEPES NaOH pH 8.0, 300 mM NaCl, 1 mM MgCl$_2$, 0.01 mM ZnSO$_4$, 0.01% IGEPAL CA-630, 10% glycerol, 1 mM PMSF, 1 mM DTT). Peak fractions were pooled, snap-frozen in liquid nitrogen, and stored at −80 °C until use.

### Reconstitution of human telomerase-DNA-TPT complex

Reconstitution of human telomerase with the DNA primer and TPT was carried out previously, as described in Sekne et al. [36]. Briefly, telomerase lysates were subjected to the oligo purification as described in *Telomerase expression and purification*. The eluate from the oligo purification was added to the pre-washed MagStrepXT resin (IBA LifeSciences, Cat# 2-4090-002) and incubated overnight at 4 °C. The resin was washed three times with wash buffer and then incubated with 2 μM telomerase DNA primer (T$_2$AG$_3$)$_5$ for 30 min at room temperature. After washing, purified TPT was added to the resin at a final concentration of 0.15 mg ml$^{-1}$ and incubated for 1 hour at 4 °C. The complex was eluted by incubation in biotin elution buffer (100 mM HEPES NaOH pH 8.0, 150 mM NaCl, 1 mM EDTA, 10 mM biotin, 2 mM MgCl$_2$, 10% glycerol, 0.1% IGEPAL CA-630, 0.2 mM PMSF and 1 mM DTT) for 1 hour at 4 °C. Fractions were analyzed on the SDS-PAGE gel followed by silver staining (Invitrogen, Cat# LC6070). The presence of the telomeric DNA substrate in the final elution was confirmed by telomerase activity assays.

### Cloning of mutant dyskerin and hTR constructs

Mutant constructs were generated as previously described[36]. Briefly, NEBaseChanger was used to design mutagenesis primers. The sequences of the primers used are listed below:

pcDNA 3.1 3xFLAG-dyskerin_I30M (Forward: TGTAGCCGAAATG-CAACACGCTG, Reverse: TCTTCTTCTGGCAATGACTTCC)

pcDNA 3.1 3xFLAG-dyskerin_Q31E (Forward: AGCCGAAATAGAA-CACGCTGAAG, Reverse: ACATCTTCTTCTGGCAATGAC)

pcDNA 3.1 3xFLAG-dyskerin_Q31K (Forward: AGCCGAAATAAAA-CACGCTGAAG, Reverse: ACATCTTCTTCTGGCAATGAC)

pcDNA 3.1 3xFLAG-dyskerin_F36V (Forward: CGCTGAAGAAGTGCT-TATCAAAC, Reverse: TGTTGTATTTCGGCTAC)

pcDNA 3.1 3xFLAG-dyskerin_K43E (Forward: ACCTGAATCCGAAGT TGCTAAGTTG, Reverse: TTGATAAGAAATTCTTCAGCGTG)

pcDNA 3.1 3xFLAG-dyskerin_L56S (Forward: GCCCCTTTTGAGCAA-GAATTTTGATAAG, Reverse: CACTGAGACGTGTCC)

pcDNA 3.1 3xFLAG-dyskerin_H68A (Forward: AAGGACAACAGCGTA TACACCTCTTGCATGTG, Reverse: ACATTCAGCTTATCAAAATTC)

pcDNA 3.1 3xFLAG-dyskerin_T66A_T67A_H68A (Forward: CGGCGTA TACACCTCTTGCATGTG, Reverse: CCGCCCTTACATTCAGCTTAT CAAAATTC)

pBS-U3-hTR-500_G450A (Forward: GCTCACACATACAGTTCCAT GG, Reverse: CGAGTCCTGGGTGCA)

pBS-U3-hTR-500_G450C (Forward: GCTCACACATCCAGTTCCAT GG, Reverse: CGAGTCCTGGGTGCA)

pBS-U3-hTR-500_G450U (Forward: GCTCACACATTCAGTTCCAT GG, Reverse: CGAGTCCTGGGTGCA)

Mutant pBS-U3-hTR-500[46] or pcDNA 3.1 3xFLAG-dyskerin[12] (Addgene, Plasmid #126870) constructs were generated using NEB Q5 Site-Directed Mutagenesis Kit (NEB, Cat# E0554) and confirmed by DNA sequencing. Plasmids were prepared using Invitrogen HiPure Plasmid Midiprep kit (Invitrogen, Cat# K210005) for transfection.

Expression and lysis experiments were carried out in triplicate. For experiments in Supplementary Fig. S9, Expi293F™ (ThermoFisher, Cat# A14527) cells were co-transfected with pcDNA 3.1-ZZ-TEV-Twin Strep-SUMOstar-TERT and wild-type or mutant pBS-U3-hTR-500 constructs. For experiments in Supplementary Fig. S7, Expi293F™ (ThermoFisher, Cat# A14527) cells were co-transfected with pcDNA 3.1-ZZ-TEV-Twin Strep-SUMOstar-TERT, pcDNA 3.1-U3-hTR-HDV and wild-type or mutant pcDNA 3.1 3xFLAG-dyskerin constructs. Cells were harvested after 48 hours and lysates were prepared by three freeze-thaw cycles.

For experiments in Supplementary Fig. S7, telomerase lysates were incubated with high-capacity streptavidin agarose resin (Thermo Scientific, Cat# 20361), prebound to a 5' biotinylated 2'-O-methyl oligonucleotide for 3 hours at room temperature[88]. The resin was washed extensively with the wash buffer (20 mM HEPES NaOH pH 8.0, 150 mM NaCl, 2 mM MgCl$_2$, 0.2 mM EGTA, 10% glycerol, 0.1% IGEPAL CA-630, 1 mM DTT and 0.2 mM PMSF) and eluted on a small column with a competitor oligonucleotide[88]. The final volume was adjusted to be equal for each mutant oligonucleotide elution (OE).

### Immunoblotting

Lysates or elution from the oligo purification (OE) from wild-type or mutant telomerase samples were resolved on a 4-12% Bis-Tris NuPAGE gel (Invitrogen, Cat# NP0321BOX) then transferred onto a nitrocellulose membrane. Membranes were blocked for 1 hour at room temperature with 5% non-fat milk, prepared in phosphate buffer saline (PBS) supplemented with 0.2% Tween-20 (PBST), followed by an overnight incubation at 4 °C with primary antibody (1:2,000 rabbit anti-dyskerin antibody, Santa Cruz Biotechnology, Cat# sc-48794, lot E0214, RRID: AB_2091314 or 1:20,000 mouse anti-alpha tubulin, ProteinTech, Cat# 66031-1-Ig, lot 10004185). The following day membranes were washed

3 times with PBST, incubated with a secondary antibody (1:10,000 goat anti-rabbit Alexa-Fluor 680, Abcam Cat# ab175773, lot GR222353-8, or 1:10,000 goat anti-mouse Alexa-Fluor 680, Abcam Cat# ab175775, lot GR3273649-2) for 1 hour at room temperature and then washed 3 additional times with PBST before imaging on a Li-COR Odyssey imager. Densitometry for quantification was performed in ImageJ.

## Telomerase activity assays of mutant telomerase

Telomerase primer extension assays were carried out as previously described[9,36]. Telomerase sample was incubated in 20-µL reactions containing 50 mM Tris-acetate pH 8.0, 4 mM MgCl$_2$, 5 mM DTT, 250 µM dTTP, 250 µM dATP, 5 µM unlabeled dGTP, 0.1 µM α-$^{32}$P-labeled dGTP (3,000 Ci mmol$^{-1}$, 10 mCi ml$^{-1}$) (Hartmann Analytic, Cat# FP-204) and 500 nM DNA reaction primer (T$_2$AG$_3$)$_5$. The reactions were performed at 30 °C for 40 min and stopped with 50 mM Tris HCl pH 7.5, 20 mM EDTA, and 0.2% SDS. DNA was extracted with phenol:chloroform:isoamyl alcohol (ThermoFisher, Cat# 17909), followed by ethanol precipitation with a $^{32}$P-labelled 18 nucleotide (nt) oligonucleotide as a recovery control (RC). Samples were resolved on a 10.5% denaturing poly-acrylamide TBE gel. The gel was dried at 80 °C for 60 min, exposed on a phosphorimager screen, and imaged using an Amersham Typhoon Biomolecular Imager (Cytiva). The telomerase activity assay was repeated three times independently. Quantification analysis was performed using ImageQuant (Cytiva), Microsoft Excel, and Prism GraphPad. The activity of the mutants was calculated as the ratio of the RC-normalized counts (total counts over the counts of the RC) and the RC-normalized counts of the wild-type. The standard error of the mean (SEM) and the pairwise one-tailed $t$-test was calculated in Microsoft Excel.

## Cryo-EM sample preparation and data collection

Cryo-EM samples were prepared previously, as described in Sekne et al. [36]. Briefly, the purified sample was crosslinked with bis(sulfosuccinimi-dyl)suberate (BS3) crosslinker (ThermoFisher, Cat# A39266) on ice for 1 hour at a final concentration of 0.5 mM BS3. The reaction was then quenched with quench buffer (200 mM Tris pH 8.0, 150 mM NaCl, 2 mM MgCl$_2$, 0.1% IGEPAL CA-630, and 1 mM DTT), followed by buffer-exchange into cryo-EM buffer (20 mM HEPES KOH pH 8.0, 150 mM NaCl, 2 mM MgCl$_2$, 0.1% IGEPAL CA-630, 1.0% trehalose and 1 mM DTT) prior to cryo-EM grid preparation. Three µl of BS3 crosslinked telomerase-DNA-TPT complex was applied onto a C-flat-T-50 4/2 grid (Protochips, Cat# CF-4/2-4Cu-T-50), pre-coated with a 5-6 nm thick layer of home-made continuous carbon film. The grids were glow-discharged using a Sputter coater discharger (model Edwards S150B). After sample application, the grids were blotted for 5 to 6 s at 4 °C and 100% humidity and vitrified in liquid ethane using an FEI Vitrobot Mark IV. Data were collected previously, as described in Sekne et al. [36]. Briefly, data collection was performed on a ThermoFisher Titan Krios transmission electron microscope operated at 300 kV and equipped with a Gatan K3 direct electron detector camera and a GIF Quantum energy filter of 20 eV slit width. EPU software was used for automatic collection in counting mode at a physical pixel size of 1.09 Å with a total electron dose of 47 to 50 electrons per Å$^2$ over an exposure time of 2.5 to 3.0 s, and a defocus range of −1 to −3 µm. Doses were fractionated into 48 movie frames. A total of 41,053 movies were collected in two separate sessions, named Dataset 1 and Dataset 2.

## Cryo-EM image processing

**H/ACA RNP lobe semi-closed dyskerin thumb loop reconstruction.** Datasets 1 and 2 were processed using RELION 4.0 (Supplementary Fig. 1)[89]. The initial processing up to signal subtraction, including motion correction, CTF estimation, particle picking, 3D classification, 2D classification, and 3D refinement was carried out previously[36]. To analyze the telomerase H/ACA RNP, we performed signal subtraction with recentering to obtain particle images of the H/ACA lobe. The box size of signal-subtracted particles was downsized to 280$^2$ pixel to speed up computation. The signal-subtracted particles were classified into eight 3D classes in two steps, with an angular sampling of 7.5°, and then 3.75°. We combined two classes with well-defined, high-resolution features to obtain a subset of 1,848,934 particles. This subset was used for 3D refinement in RELION and also further heterogeneity analyzes in cryoSPARC[71,90] (see below).

The 3D refinement in RELION resulted in a 3.2 Å reconstruction. The angular assignments from this RELION refinement were used for alignment-free 3D classification with a regularization parameter $T$ of 12. The best 3D class with 293,764 particles was refined to 2.9 Å resolution. CTF refinement (beam tilt, trefoil, and 4$^{th}$ order aberrations, anisotropic magnification, per-particle defocus, and per-micrograph astigmatism)[91] improved the reconstruction to 2.7 Å resolution. These signal-subtracted particles were subsequently reverted to the original particles for Bayesian polishing[92]. The polished particles were refined and signal-subtracted with recentering to remove the signal from the catalytic core. The signal-subtracted particles were refined to 2.8 Å resolution. We performed CTF refinement (beam tilt, trefoil, and 4$^{th}$ order aberrations, anisotropic magnification, per-particle defocus, and per-micrograph astigmatism), followed by 3D refinement (2.8 Å resolution). Although the resolution is unchanged, CTF refinement improves the quality of the map. To further improve particle homogeneity, we performed another round of alignment-free 3D classification with a regularization parameter $T$ of 14. A subset of 240,590 particles with the most intact density was selected for 3D refinement, resulting in a 2.7 Å reconstruction (Fig. 1d, e).

To resolve the 3′ dyskerin thumb loop, a mask that included only the regions around the thumb loop was used for focused alignment-free 3D classification with a regularization parameter $T$ of 500. A subset of 199,360 particles, which showed the most well-resolved density of the thumb loop, was refined to 2.7 Å resolution. This consensus reconstruction was used for model building, refinement, and validation.

**H/ACA RNP lobe open dyskerin thumb loop reconstruction.** A subset of 1,848,934 particles was exported as a particle stack from RELION for processing in CryoSPARC v4.1.2 with a corrected pixel size of 1.059 Å (Supplementary Fig. 11)[90]. All steps were carried out using the default parameters unless stated otherwise. Non-uniform refinement[93] was performed using a consensus reconstruction from RELION as the initial volume. Refinement of per-particle defocus, beam tilt, trefoil, spherical aberration, and anisotropic magnification resulted in a 3.8 Å reconstruction. The particles were then classified into six volumes using the heterogeneous refinement algorithm[90] with the non-uniform refined map as an initial model. The three best classes with 1,196,982 particles were subjected to non-uniform refinement, followed by CTF refinement to yield a 2.8 Å reconstruction. 3D-variability analysis was then performed by selecting for 4 modes, using a low-pass filter resolution of 5 Å and a mask that excluded the highly flexible parts of hTR P4 (Fig. 1c and Supplementary Fig. 3a, b). This analysis uncovered the variability of the 3′ dyskerin thumb loop (Fig. 8 and Supplementary Fig. 11). Therefore, the non-uniform refinement was subjected to focused 3D classification into 10 classes, using a mask around the thumb loop-P7/P8a RNA junction (Fig. 1c and Supplementary Fig. 3a, b), a target resolution of 5 Å and with hard classification. We identified classes with the thumb loop in the open conformation. The most populated class of 145,627 particles with the best resolved density was subsequently refined to 3.1 Å resolution using non-uniform refinement (Supplementary Fig. 11a). The particles were exported as a particle stack using PyEM[94] for further processing in RELION 4.0. In RELION, we refined this particle stack and performed CTF refinement (beam tilt, trefoil, and 4$^{th}$ order aberrations, anisotropic magnification and per-particle defocus), followed by 3D refinement to yield a final 3.1 Å reconstruction.

For both maps, 3D refinement was performed using fully independent data half-sets. Reported resolution was based on the

gold-standard Fourier shell correlation (FSC) = 0.143 criterion between the two half-maps (Supplementary Fig. 2, 11)[95,96], with FSCs calculated with a soft mask. The pixel size was corrected to 1.059 Å during post-processing and CTF refinement in RELION or during import into CryoSPARC. The map was corrected for the modulation transfer function of the detector and sharpened by applying a negative B-factor. B-factors were determined by RELION or as a user-defined value (also see Supplementary Table 1). Local resolution was calculated within RELION (Supplementary Fig. 2, 11). 2D histograms of the Euler angles covered by refined particles were calculated using a Python script (https://githubhelp.com/Guillawme/angdist).

### Model building, refinement and AlphaFold2 prediction

We used the published atomic model of the H/ACA RNP lobe (PDB 7BGB [https://doi.org/10.2210/pdb7BGB/pdb])[10] as an initial model for model building. Newly resolved regions (Supplementary Fig. 5) were built in COOT 0.9.8.1[97], either de novo or using AlphaFold2 models as guides[58]. To facilitate model building, we converted all maps from MRC format into MTZ format using REFMAC5.8[98,99] to allow map blurring and sharpening in COOT. The model was then refitted into the map in ISOLDE[100]. RNA geometry was improved using the ERRASER ROSIE webserver (https://rosie.graylab.jhu.edu/erraser/)[101,102]. To improve the fit of the RNA in the cryo-EM map, we used ISOLDE with adaptive distance restraints to maintain base-pairing[103]. Before model refinement, ISOLDE and COOT were iteratively used to diagnose and fix errors, and to improve model geometry.

We used the resulting model, described above, to build an atomic model into the map with the dyskerin thumb loop resolved in an open conformation (Fig. 8 and Supplementary Fig. 11, 12). The fit was first improved in ISOLDE with adaptive distance and torsion restraints for low resolution regions at the periphery of the density. Regions that underwent conformational changes, such as the hTR P7/P8a junction and the 3' dyskerin thumb loop, were manually rebuilt in COOT. RNA and protein model geometry was improved as described above.

For the consensus model, model refinement was performed in REFMAC5.8 in reciprocal space with protein secondary structure restraints and nucleic acid restraints calculated using PROSMART[104] and LIBG[99], respectively. For the model with the dyskerin thumb loop in the open conformation, model refinement was first performed in real space using PHENIX 1.20[105] (one macro cycle of global minimization) followed by refinement using REFMAC5.8 as described above. For both models, Model-vs-map FSCs and EMRinger scores were calculated using Phenix 1.20[105]. Q-scores[106] were calculated in UCSF Chimera and included in Supplementary Data 3 and 4[107]. Geometries were assessed using the MolProbity server (http://molprobity.biochem.duke.edu/)[108]. Supplementary Table 1 provides a summary of the refined models. Pymol sessions of the refined models are also included as Supplementary Data 1 and 2 in Additional Supplementary files.

AlphaFold2 was used for structure prediction of the H/ACA RNP as well as human dyskerin-SHQ1 complex[58]. The top ranked model was used as a template for model building (H/ACA RNP) and for interaction analysis (dyskerin-SHQ1 complex) (Supplementary Fig. 8).

### Map and model visualization

Maps were visualized with UCSF Chimera[107] and UCSF ChimeraX[109]. Illustrations were prepared using Adobe Illustrator, Chimera, ChimeraX and Pymol (www.pymol.org).

### Reporting summary

Further information on research design is available in the Nature Portfolio Reporting Summary linked to this article.

## Data availability

The cryo-EM maps of the telomerase H/ACA RNP semi-closed state and open state have been deposited in the Electron Microscopy Database (EMDB) under accession codes EMD-17190 and EMD-17191, respectively. PDB coordinates for the telomerase H/ACA RNP lobe semi-closed state and open state have been deposited in the Protein Data Bank under accession codes 8OUE and 8OUF, respectively. Pymol sessions of the deposited PDB models are also included in Supplementary Data 1 and 2 in Additional Supplementary files. The source data underlying Supplementary Fig. 7a, b and Supplementary Fig. 9a are provided as a Source Data file. Source data are provided with this paper.

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

## Acknowledgements

We thank the MRC LMB EM facility staff for access and support of EM sample preparation and data collection; J. Grimmett, T. Darling and I. Clayson for maintaining the Scientific Computing facility; Scheres lab for RELION data processing advice; S. Thorkelsson for cryoSPARC data processing advice; the Nagai, Löwe, Passmore, and Scheres labs for sharing reagents, equipment, and technical advice; L. Passmore, V.

Chandrasekaran and S. Thorkelsson for critical reading of the manuscript. This work was supported by the Medical Research Council, as part of United Kingdom Research and Innovation (also known as UK Research and Innovation) [MC_UP_1201/19]. For the purpose of open access, the MRC Laboratory of Molecular Biology has applied a CC BY public copyright license to any Author Accepted Manuscript version arising. *Funding:* UKRI-Medical Research Council grant MC_UP_1201/19 (T.H.D.N.). EMBO Young Investigator Award (T.H.D.N). Jane Coffin Childs Postdoctoral Fellowship (G.E.G.).

## Author contributions

Z.S. and G.E.G. analyzed EM data and performed model fitting. Z.S. performed biochemical experiments and quantification. G.E.G. performed model building with Z.S. G.E.G. and Z.S. performed model refinement. A.-M.M.v.R. prepared telomerase extracts and performed initial model refinements. S.B. prepared telomerase expression constructs. Z.S., G.E.G., and T.H.D.N. analyzed the structures. Z.S. and G.E.G. prepared the first draft of the paper. Z.S., G.E.G., and T.H.D.N. finalized the paper with feedback from all authors.

## Competing interests

The authors declare no competing interests.
