## [Peer Review File · Nature Communications]

2.7 Å cryo-EM structure of human telomerase H/ACA ribonucleoproteinREVIEWER COMMENTS

Reviewer #1 (Remarks to the Author):

Summary:

In this manuscript, Ghanim et al. report a 2.7 Å cryo-EM structure of the H/ACA Ribonucleoprotein (RNP) of the human telomerase. The improved map resolution enables them to inspect and analyze molecular interactions governing telomerase H/ACA RNP assembly. In addition, from the telomerase H/ACA RNP cryo-EM structure, they gained new insights into the pseudouridylation mechanism of canonical eukaryotic H/ACA RNPs. These findings are derived from new cryo-EM analysis performed on existing datasets that are published by the authors. Before I support the publication of this manuscript, I have several concerns that I believe the authors should address. Because these concerns impact the authors interpretation and main conclusions. Please see the following comments for details:

Major comments:

1. While the quality of the cryo-EM map of the telomerase H/ACA RNP has improved (amazing effort in sorting the complex heterogeneity!), allowing the authors to better identify key interactions underpinning the RNP assembly, most of the author's analysis are limited to hand-waving interpretations without further biochemical validation. I appreciate the connections of these identified interaction sites to human disease mutations, but I wonder if the authors have attempted to validate some of the new interactions that they found and assign them to a particular function for telomerase biogenesis?
2. Many of the analyses do not provide details to the interactions that the authors identified. For example, the main text lacks description of the interactions that the authors identified between the NTE domain and helix 1 (Figure 2); page 9 line 273, "5' dyskerin via numerous interactions..". What are these interactions? Which residue to which residue, and what kind of interaction? I strongly encourage the authors to provide these details in the main text for all new interactions that they have identified and described in the manuscript.
3. For sections describing dyskerin-hTR and TCAB1-hTR-NHP2 interactions, the authors should elaborate how their findings compare with those done in Wan et al., 2021.
4. Page 10, line 275-276: The authors should elaborate how would a F36V or K43E mutation disrupts the inter-dyskerin interactions. Are these residues conserved? A conservation analysis on the residues involved in the interaction could be helpful; same applies to Q31 in page 10, line 283.
5. Page 10, line 299: This passage could be confusing to readers new to the telomerase H/ACA RNP structure. Perhaps, a cartoon model showing why the NOP10 interaction with the 5' dyskerin can strengthen inter-dyskerin interactions.
6. Page 11, line 308: It will be helpful to overlay these disease mutation residues on the HBM in a relevant figure panel.
7. Page 11, line 313: The authors should describe the "same interaction" in more details. Figure S7c-e, residues are drawn but it is unclear which residue interact with which one.
8. Page 11, line 314: "Mutations in SHQ1...". Which mutation residues are they? Please include these information in this sentence and highlight them in the corresponding figures.
9. Page 11, line 323-326: The G450 hTR residue participates in the H/ACA lobe of telomerase and not in the catalytic domain. What would be the rationale in testing G450 base mutations using telomerase activity assays? As the authors pointed out, hTR maturation or telomerase assembly is likely impacted or the assembly of the telomerase in cells. Have the authors tested if these mutations affect telomerase recruitment to telomeres/Cajal bodies in human cultured cells? It would be helpful (for non-telomerase field readers) to explain briefly why the authors choice of reconstitution system would bypass hTR 3' end processing.
10. Page 11, line 334 and page 12, line 341: Residues 317-326 are mentioned but which of these residues or all are involved in the contact with the P8 stem loop? What are the interactions between TCAB1 and 3' NHP2?

Minor comments:

1. Abstract: line 17 – might be helpful to include the range of resolution for the mentioned previous structures. This helps readers understand the resolution gap between this manuscript work and past structures.
2. Figure 2: To prevent confusion, perhaps the authors should mention that only the DKC1 subunits are illustrated in the figure legend.
3. Figure panel 2C: The authors claimed a hydrophobic cleft formed by the 3' NTE. The authors should consider showing this analysis as a surface representation using hydrophobicity or electrostatic potential coloring in a new figure panel or supplementary data.
4. Figure S6: Resolution is poor. A higher DPI figure will be needed.
5. Page 10, line 299: The authors should cite references or data to “the disease mutations” that disrupt the observed interactions. It will be great for the readers if the authors can depict these mutations in Figure 2 or 3.
6. Recommend moving figure panel 4D to 4C to help reader follow the flow of the main text.
7. Page 11, line 319: “at the 3' end of hTR (G45C) (Fig. 4c)”. Should Figure S4e be cited too?
8. Page 12, line 355: “a well-resolved thumb loop for 3' dyskerin”. Is there map over model figure for this loop?
9. Page 12, line 362-363: “flips out towards the dyskerin active site”. Is this supposed to be “flips out away”?
10. Page 12, line 363: Define dyskerin active site residue(s).
11. Page 12, line 376-377: If the human GAR1 helix1 are analogous to the yeast one, are there any sequence conservation between their sequences?
12. Page 15, line 446: How would the authors envision a small molecule stabilizes the inter-dyskerin interface?

Reviewer #2 (Remarks to the Author):

Ghanim et al. provided the highest resolution cryo-EM structure of human telomerase so far. Here, they take advantage of that structure to further resolve the structural H/ACA lobe of telomerase that is required for biogenesis of telomerase and stability of the human telomerase RNA (hTR). Human telomerase consists of two protein heterotetramers, 5' and 3', the Cajal body localizing protein TCAB1, the reverse transcriptase TERT, and histones H2A and H2B, all assembled on hTR. The H/ACA lobe is formed by the two heterotetramers, each consisting of one dyskerin, NOP10, NHP2, and a GAR1 subunit bound to one of the two hTR 5' and 3' hairpins. TCAB1 binds to the 3' hairpin. The H/ACA lobe is of interest because it not only forms part of telomerase but is part of hundreds of H/ACA RNPs involved in ribosomal and spliceosomal RNA pseudouridylation. Importantly, the proteins of the H/ACA lobe harbor many of the missense mutations causing short telomeres and the bone marrow failure syndrome dyskeratosis congenita. The 2.7 Å high-resolution structure now reveals several additional amino acids and nucleotides of the H/ACA lobe explaining some of the impact of dyskeratosis congenita (DC) mutations on the telomerase complex.

The asymmetric interaction of the two dyskerins (identical in amino acid sequence) is described in detail. The interaction between hTR and the two dyskerins is outlined. The binding of TCAB1, not only to hTR, but also the 3' NHP2 is detailed. The thumb loop domain of dyskerin, important for substrate binding and release in archaeal ACA RNPs, is resolved in two conformations explaining not only substrate binding of eukaryal guide RNAs but also that of hTR. Thus, part 7 of hTR forms a pseudosubstrate that is held in place by the thumb loop like the hybrid of the pseudouridylation pocket and target RNA. Altogether, these are important refinements to the structure that give insight into disease mechanism and pseudouridylation H/ACA RNPs. The manuscript is very clearly written and illustrated. I have just some general questions.

1. A major conclusion seems to be that the 5' and 3' half of the H/ACA lobe differ, i.e., the 3'

resembles a regular pseudouridylation sno/scaRNP. In contrast, the 5' half differs from the pseudouridylation particle because the 5' hairpin of hTR twists away from the dyskerin-NOP10-NHP2 axis. In pseudouridylation H/ACA RNPs, however, the 5' hairpin must be similarly aligned with the three-protein axis as the 3' hairpin to place the pseudouridylation pocket near the catalytic aspartate. So, on one hand the difference in structure between the telomerase RNP and the pseudouridylation RNPs could explain a potential different impact of DC mutations on telomerase than the other RNPs. Yet, in both heterotetramers, DC mutations are identical making it difficult to understand the apparently diverse impact of the mutations.

2. Similarly, the model does not explain the proven pseudouridylation activity of some 5' hairpins. Is the structure of the telomerase RNP indeed different from that of pseudouridylation guide RNPs? Perhaps, mentioning of this issue and some speculation would help.

3. The asymmetric binding of the two identical dyskerins is well explained and illustrated. How can two proteins identical in amino acid sequence form such an asymmetric bond? Is this prion-like? And if so, why do they not form multimers? Especially, it might be enlightening to mention if there is any precedent for such a case.

4. It seems interesting that the pseudosubstrate conformation of hTR aids its association with the H/ACA protein complex (Fig. 7). In the case of snoRNPs, however, snoRNAs easily associate with the protein complex in the absence of substrate RNA. Thus, the pseudosubstrate conformation may not be necessary?

5. On the other hand, the hTR pseudosubstrate could explain why an artificial substrate matching that sequence is not pseudouridylated unlike those matching other H/ACA RNAs.

6. Although the model explains the impact of dyskerin DC mutations on the NOP10-dyskerin interaction, it does not address the impact of the R34W NOP10 DC mutation.

7. There is no mention, that the long intrinsically disordered N- and C-terminal tails of dyskerin and GAR1 are invisible in the structure. These tails could impact some of the interpretations reached in the manuscript.

8. Finally, it is not clear what "circle" refers to in Fig. 2b on p.9, line 268?

Tom Meier

Reviewer #3 (Remarks to the Author):

Comments to the Author

The discovery of telomerase almost forty years ago highlighted its significance as a potential target for anti-cancer and aging treatments. However, the lack of comprehensive structural information has impeded the progress in developing drugs against telomerase. This research addresses this issue by improving the data processing procedures of an existing dataset, resulting in the determination of a 2.7 Å resolution structure of the H/ACA RNP lobe of human telomerase. This level of resolution enables the identification of intricate molecular interactions within the H/ACA RNP, the mapping of disease-related mutations, and the potential guidance for structure-based drug design. Intriguingly, although not yet confirmed, this study proposes that human telomerase possesses pseudouridylation activity based on the observation of two conformations of the 3' dyskerin thumb loop. However, several questions remain to be answered before this research can be published.

1. In methods, please combine the two parts: "Model building and refinement" and "AlphaFold2 prediction".
2. This study addresses the high resolutions of the two maps of the telomerase H/ACA RNP: semi-closed state and open state. Further evaluation techniques are needed, like applying the Q-score to demonstrate and confirm the resolvability of individual hTR and protein subunits' residues, rather than merely providing an overall average value in the table.
3. Line 253-255, delete it or simply summarize its results.
4. To gain insights into the pathogenic mechanisms of mutations, certain functional assays must be conducted. These assays aim to investigate the impact of mutations on factors such as enzyme assembly, activity, and other relevant aspects.
5. Line 293: a little confusing, need to clarify the direct interaction.
6. Line 310, AlphaFold prediction can be done to verify it.
7. The two parts "Consensus map reveals a non-catalytic semi-closed conformation of the 3' dyskerin thumb loop" and "hTR mimics the substrate-guide RNA duplex of a H/ACA RNP" can be combined to prevent unnecessary duplication.
8. Line 406: a little controversial with the above mentioned "a well-resolved thumb", and please highlight it in the figure.
9. The authors have postulated the potential presence of pseudouridylation activity in telomerase. Nonetheless, there remains a question regarding the mechanism by which hTR switches with substrate RNA.
10. To substantiate the authors' proposal, it is imperative to conduct an in vitro assay to assess the pseudouridylation activity of telomerase.

REVIEWER COMMENTS

We would like to thank the reviewers their careful reading of the manuscript and their constructive comments to improve the work. We have studied each point raised by the reviewers and revised the manuscript accordingly. We also conducted additional experiments to validate the new interactions described in the manuscript. Below is our detailed point-by-point response to each of the reviewers' comments.

Reviewer #1 (Remarks to the Author):

Summary:

In this manuscript, Ghanim et al. report a 2.7 Å cryo-EM structure of the H/ACA Ribonucleoprotein (RNP) of the human telomerase. The improved map resolution enables them to inspect and analyze molecular interactions governing telomerase H/ACA RNP assembly. In addition, from the telomerase H/ACA RNP cryo-EM structure, they gained new insights into the pseudouridylation mechanism of canonical eukaryotic H/ACA RNPs. These findings are derived from new cryo-EM analysis performed on existing datasets that are published by the authors. Before I support the publication of this manuscript, I have several concerns that I believe the authors should address. Because these concerns impact the authors interpretation and main conclusions. Please see the following comments for details:

We thank the reviewers for the positive remarks on the manuscript.

Major comments:

1. While the quality of the cryo-EM map of the telomerase H/ACA RNP has improved (amazing effort in sorting the complex heterogeneity!), allowing the authors to better identify key interactions underpinning the RNP assembly, most of the author's analysis are limited to hand-waving interpretations without further biochemical validation. I appreciate the connections of these identified interaction sites to human disease mutations, but I wonder if the authors have attempted to validate some of the new interactions that they found and assign them to a particular function for telomerase biogenesis?

We agree with the reviewer's point. However, the reconstitution approach used to prepare telomerase for biochemical and structural studies in this study and also in the telomerase field in general limits our ability to perform biochemical validation. To reconstitute telomerase, we overexpressed only TERT and hTR in human cells. Therefore, we rely on endogenous H/ACA components, which are present in a vast cellular excess compared to TERT and hTR, to assemble with the overexpressed TERT and hTR to form telomerase holoenzyme. We also used a hepatitis delta virus (HDV) ribozyme at the 3' end of hTR and thus bypass the cellular 3' end processing of hTR. Under these conditions, performing *in vivo* mutagenesis properly will require endogenous knockout of telomerase components, which is lethal for cells because of their involvement in highly active ribosome and spliceosome biogenesis pathways. Likewise, knockdown of H/ACA RNP components will also affect cell survival. Furthermore, to obtain sufficient telomerase for biochemical characterisation, we transfect a large number of plates of cells. At such scale, knocking down dyskerin would be challenging. In addition, to our current knowledge, telomerase biogenesis and the 3' end processing of hTR are active areas of research and not very well-understood. Therefore, even if we could generate mutants, the tools to dissect the effects of the mutants are well beyond our expertise and outside the scope of this manuscript.

However, to address the reviewer's comment, we performed the best possible experiments that our current experimental setup allows, and have described them in detail below.

Regarding interactions involving dyskerin, we prepared overexpression constructs of Flag-tagged wild-type and mutant dyskerin. We selected 8 disease mutations in dyskerin at residues involved in key dyskerin-dyskerin and dyskerin-hTR interactions described in the manuscript, including I30M, Q31E, Q31K, F36V, K43E, L56S, H68A and T66A/T67A/H68A. We transfected these dyskerin constructs together with the TERT and hTR constructs. As mentioned above, endogenous dyskerin was still present in these experiments. We used Western blot to monitor the expression levels of TERT and Flag-tagged dyskerin and also the level of endogenous dyskerin in the lysate (Supplementary Fig. 7a). Although the same amounts of Flag-tagged dyskerin DNA were used for transfection, the expression levels of all mutant dyskerin were lower than the Flag-tagged wildtype dyskerin (Supplementary Fig. 7a). It is possible that these mutations cause instability in dyskerin. We then performed oligo-based purification on hTR. We observed that the overexpressed Flag-tagged wild-type dyskerin efficiently outcompeted endogenous dyskerin for incorporation into telomerase, evident in the high Flag-dyskerin/total dyskerin ratio in the Western blot (Supplementary Fig. 7b and c). In contrast, all dyskerin mutants competed less efficiently with the endogenous dyskerin for incorporation into telomerase. This is likely caused by the weakened inter-dyskerin or dyskerin-hTR interactions in the dyskerin mutants, validating the interactions we observed in our structures. We performed these

experiments in three independent replicates and our observations were consistent across the replicates (Supplementary Fig. 7).

To avoid bypassing 3' end processing of hTR, we used a different hTR expression construct (pBS-U3-hTR-500), in which the HDV ribozyme sequence is replaced with 500 nucleotides of the endogenous 3' UTR of hTR (Fu et al., 2003). To validate the interactions involving nucleotide G450 of hTR, we reconstituted telomerase with G450A, G450C and G450U mutations and performed activity assays (Supplementary Fig. 9). Unlike the experiments included in the submitted manuscript, these assays showed that these mutations resulted in a slight but significant reduction of telomerase activity. In our model, G450 stacks with H68 of the 3' dyskerin and hydrogen bonds with S42 of the 5' dyskerin. The effects are not large because all alternative nucleotides can maintain the stacking interactions with H68 to some extent. The results are consistent across three independent replicates. Mutating G450 to A maintains the purine-H68 stacking while disrupting the hydrogen bond with S42. On the other hand, mutating G450 to a pyrimidine (C or U) reduces the stacking interaction with H68 and disrupts the hydrogen bond with S42. Therefore, the reductions in telomerase activity we observed in G450C and G450U were more pronounced than that in G450A.

We have incorporated these new experiments in the revised manuscript (lines 106-137, lines 316-326 and 368-384).

2. Many of the analyses do not provide details to the interactions that the authors identified. For example, the main text lacks description of the interactions that the authors identified between the NTE domain and helix 1 (Figure 2); page 9 line 273, "5' dyskerin via numerous interactions..". What are these interactions? Which residue to which residue, and what kind of interaction? I strongly encourage the authors to provide these details in the main text for all new interactions that they have identified and described in the manuscript.

We completely understand the reviewer's point. We initially simplified the manuscript to avoid long descriptions of interactions for clarity. We have now added more details about the mentioned interactions in the revised version of the manuscript (lines 297-301).

3. For sections describing dyskerin-hTR and TCAB1-hTR-NHP2 interactions, the authors should elaborate how their findings compare with those done in Wan et al., 2021.

We agree with the reviewer that we may not have clearly stated this comparison in the submitted version of the manuscript. The interactions we highlight in our manuscript are from newly resolved regions of hTR, dyskerin and TCAB1 compared to our published structure and that of Wan et al., 2021. For dyskerin-hTR interactions, we have mentioned comparisons to previous structures in the manuscript (lines 345-347). We have now added additional texts to emphasize these points for TCAB1 (lines 387-390).

4. Page 10, line 275-276: The authors should elaborate how would a F36V or K43E mutation disrupts the inter-dyskerin interactions. Are these residues conserved? A conservation analysis on the residues involved in the interaction could be helpful; same applies to Q31 in page 10, line 283.

We have now elaborated the effects of the F36V, K43E and Q31E mutations and added the description of Q31 in the revised manuscript (line 305, line 312). Both F36, K43 and Q31 are highly conserved across a range of species as shown in Supplementary Figure 6. We have modified this figure to highlight these residues.

5. Page 10, line 299: This passage could be confusing to readers new to the telomerase H/ACA RNP structure. Perhaps, a cartoon model showing why the NOP10 interaction with the 5' dyskerin can strengthen inter-dyskerin interactions.

This is a helpful suggestion from the reviewer. We have modified the text to refer to Figure 3a (line 341), which shows the interactions schematically.

6. Page 11, line 308: It will be helpful to overlay these disease mutation residues on the HBM in a relevant figure panel.

In Figure 4d, the disease mutations on the HBM are highlighted as spheres.

7. Page 11, line 313: The authors should describe the "same interaction" in more details. Figure S7c-e, residues are drawn but it is unclear which residue interact with which one.

We have changed the main text, which now reads "AlphaFold2 predicts positioning of the human HBM domain on the SHQ1 in a similar manner as observed in yeast (Supplementary Fig. 8c-e)" (lines 355-356). We have modified Supplementary Figure 8d to highlight the interactions. Please also refer to point 8.

8. Page 11, line 314: “Mutations in SHQ1...”. Which mutation residues are they? Please include these information in this sentence and highlight them in the corresponding figures.

We thank the reviewer for the suggestion. We have revised the text to include an example of a residue A426 (line 356), modified Supplementary Figure 8d to show the mutant residues on both SHQ1 and HBM, and indicated the interactions between them.

9. Page 11, line 323-326: The G450 hTR residue participates in the H/ACA lobe of telomerase and not in the catalytic domain. What would be the rationale in testing G450 base mutations using telomerase activity assays? As the authors pointed out, hTR maturation or telomerase assembly is likely impacted or the assembly of the telomerase in cells. Have the authors tested if these mutations affect telomerase recruitment to telomeres/Cajal bodies in human cultured cells? It would be helpful (for non-telomerase field readers) to explain briefly why the authors choice of reconstitution system would bypass hTR 3' end processing.

We agree with the reviewer on this point. However, as explained in our answer to Point 1, such *in vivo* experiments are very challenging to perform and outside of the scope of the paper. We had hoped that the G450 base mutation would sufficiently disrupt telomerase assembly, and manifest as a decrease in telomerase activity, irrespective of the reconstitution system we used. However, this does not seem to be the case. Instead, we performed additional experiments to validate the interactions between G450 and dyskerin as described in detail in our response to Point 1.

10. Page 11, line 334 and page 12, line 341: Residues 317-326 are mentioned but which of these residues or all are involved in the contact with the P8 stem loop? What are the interactions between TCAB1 and 3' NHP2?

We have modified the text to provide more details of the interactions between TCAB1 loop 317-326 and P8 stem loop (line 391). We also added a more detailed description for the interaction between TCAB1 and 3' NHP2 in the revised text (lines 399-401).

Minor comments:

1. Abstract: line 17 – might be helpful to include the range of resolution for the mentioned previous structures. This helps readers understand the resolution gap between this manuscript work and past structures.

This is a very helpful suggestion. We have added the resolution range of the previous structures in the main text (line 17).

2. Figure 2: To prevent confusion, perhaps the authors should mention that only the DKC1 subunits are illustrated in the figure legend.

We have now added the following sentence to the figure legend of Figure 2 to improve clarity as suggested by the reviewer.

3. Figure panel 2C: The authors claimed a hydrophobic cleft formed by the 3' NTE. The authors should consider showing this analysis as a surface representation using hydrophobicity or electrostatic potential coloring in a new figure panel or supplementary data.

We have modified Figure 2 as suggested by the reviewer.

4. Figure S6: Resolution is poor. A higher DPI figure will be needed.

This was an oversight. We thank the reviewer for noting this. We have replaced Supplementary Figure 6 with a higher resolution figure.

5. Page 10, line 299: The authors should cite references or data to “the disease mutations” that disrupt the observed interactions. It will be great for the readers if the authors can depict these mutations in Figure 2 or 3.

We have now added citations for the disease mutations (line 343) and modified Figure 2g to depict these mutations.

6. Recommend moving figure panel 4D to 4C to help reader follow the flow of the main text.

The original panel organization in Figure 4 was purposeful to allow for comparisons between the interacting residues and location of the disease mutations. However, we understand the reviewer's point and have rearranged the panels in Figure 4 to better follow the flow of the main text.

7. Page 11, line 319: "at the 3' end of hTR (G45C) (Fig. 4c)". Should Figure S4e be cited too?

We thank the reviewer for noting this. We have changed the text to reflect this suggestion (line 362).

8. Page 12, line 355: "a well-resolved thumb loop for 3' dyskerin". Is there map over model figure for this loop?

Thank you for noting this. A map over model figure is included in Figure 8c and Supplementary Figure 3e, inset. We have now included a reference to Supplementary Figure 3e.

9. Page 12, line 362-363: "flips out towards the dyskerin active site". Is this supposed to be "flips out away"?

We may not have been clear in the text. hTR nucleotide G393 flips out of the RNA duplex towards the dyskerin active site. We have modified the text to clarify this point as followed " Nucleotide G393 of hTR flips out of the RNA duplex..." (line 419).

10. Page 12, line 363: Define dyskerin active site residue(s).

We have now modified the text to define the dyskerin active site residue (lines 405-406).

11. Page 12, line 376-377: If the human GAR1 helix1 are analogous to the yeast one, are there any sequence conservation between their sequences?

This is a good point raised by the reviewer. There is indeed strong sequence conservation in GAR1 helix1 across several species, including yeast. Supplementary Figure 6b shows sequence alignments. We have added helix 1 label in this figure to highlight this point and make it easier for the readers to see.

12. Page 15, line 446: How would the authors envision a small molecule stabilizes the inter-dyskerin interface?

We envision small molecules that stabilize the inter-dyskerin interface based on the concepts reviewed in Fischer et al., 2016 (10.1016/j.sbi.2016.01.004) and Bier et al. 2016 (10.1002/cmdc.201500484). Such small molecules can be designed to bind the dyskerin-dyskerin interface and provide additional contacts between the subunits that would stabilize their interactions. It is also possible to design a small molecule to act as a "molecular glue" as described in Fischer et al., 2016 and overcome the destabilizing effects of some disease mutations.

Reviewer #2 (Remarks to the Author):

Ghanim et al. provided the highest resolution cryo-EM structure of human telomerase so far. Here, they take advantage of that structure to further resolve the structural H/ACA lobe of telomerase that is required for biogenesis of telomerase and stability of the human telomerase RNA (hTR). Human telomerase consists of two protein heterotetramers, 5' and 3', the Cajal body localizing protein TCAB1, the reverse transcriptase TERT, and histones H2A and H2B, all assembled on hTR. The H/ACA lobe is formed by the two heterotetramers, each consisting of one dyskerin, NOP10, NHP2, and a GAR1 subunit bound to one of the two hTR 5' and 3' hairpins. TCAB1 binds to the 3' hairpin. The H/ACA lobe is of interest because it not only forms part of telomerase but is part of hundreds of H/ACA RNPs involved in ribosomal and spliceosomal RNA pseudouridylation. Importantly, the proteins of the H/ACA lobe harbor many of the missense mutations causing short telomeres and the bone marrow failure syndrome dyskeratosis congenita. The 2.7 Å high-resolution structure now reveals several additional amino acids and nucleotides of the H/ACA lobe explaining some of the impact of dyskeratosis congenita (DC) mutations on the telomerase complex.

The asymmetric interaction of the two dyskerins (identical in amino acid sequence) is described in detail. The interaction between hTR and the two dyskerins is outlined. The binding of TCAB1, not only to hTR, but also the 3' NHP2 is detailed. The thumb loop domain of dyskerin, important for substrate binding and release in archaeal ACA RNPs, is resolved in two conformations explaining not only substrate binding of eukaryal guide RNAs but also that of hTR. Thus, part 7 of hTR forms a pseudosubstrate that is held in place by the thumb loop like the hybrid of the pseudouridylation pocket and target RNA. Altogether, these are important refinements to the structure that give insight into disease mechanism and pseudouridylation H/ACA RNPs. The manuscript is very clearly written and illustrated. I have just some general questions.

We thank the reviewer for the positive remarks on the findings presented in this manuscript.

1. A major conclusion seems to be that the 5' and 3' half of the H/ACA lobe differ, i.e., the 3' resembles a regular

pseudouridylation sno/scaRNP. In contrast, the 5' half differs from the pseudouridylation particle because the 5' hairpin of hTR twists away from the dyskerin-NOP10-NHP2 axis. In pseudouridylation H/ACA RNPs, however, the 5' hairpin must be similarly aligned with the three-protein axis as the 3' hairpin to place the pseudouridylation pocket near the catalytic aspartate. So, on one hand the difference in structure between the telomerase RNP and the pseudouridylation RNPs could explain a potential different impact of DC mutations on telomerase than the other RNPs. Yet, in both heterotetramers, DC mutations are identical making it difficult to understand the apparently diverse impact of the mutations.

We understand the reviewer's concern. What the reviewer stated is indeed true. Although the DC mutations are identical in both heterotetramers, the two dyskerin molecules do not make identical interactions with hTR and each other in telomerase. When mapping the DC mutations in both heterotetramers, the majority cluster at the interface between the dyskerin subunits (Nguyen, et al 2018), disrupting not just one, but reciprocal interactions between them (Ghanim, et al 2021). In telomerase, the 5' hairpin of hTR is noncanonical and makes fewer interactions with the 5' H/ACA heterotetramer. This loss of RNA-protein interactions makes 5' hairpin assembly with the 5' heterotetramer more dependant of cross-hairpin dyskerin-dyskerin interactions. Taken together, this rationalizes why DC mutations have a more severe impact on telomerase than other H/ACA RNPs.

2. Similarly, the model does not explain the proven pseudouridylation activity of some 5' hairpins. Is the structure of the telomerase RNP indeed different from that of pseudouridylation guide RNPs? Perhaps, mentioning of this issue and some speculation would help.

This is an excellent point by the reviewer. We anticipate that the structure of telomerase H/ACA RNP and the other H/ACA RNPs would be similar, except for the lack of additional RNA interactions in the 5' heterotetramer of telomerase. Although the thumb loop of the telomerase 5' dyskerin was unresolved in our structure, we cannot rule out that the 5' hairpin could still possess pseudouridylation activity in other H/ACA RNPs. Interestingly, the CTE helix of the 5' GAR1 is in a similar position relative to the putative 5' dyskerin active site compared to the 3' counterpart. Overlaying the 3' dyskerin over the 5' dyskerin shows that the region of the 5' dyskerin thumb loop faces the solvent. It is possible that the weaker interactions with hTR result in a more mobile 5' dyskerin thumb loop. However, it is too speculative for us to make a conclusion on this. We now add a paragraph to the manuscript (lines 435-439) to comment on this point.

3. The asymmetric binding of the two identical dyskerins is well explained and illustrated. How can two proteins identical in amino acid sequence form such an asymmetric bond? Is this prion-like? And if so, why do they not form multimers? Especially, it might be enlightening to mention if there is any precedent for such a case.

We agree with the reviewer that this is a fascinating aspect of the inter-dyskerin interactions. The observed interactions do not appear to be prion-like. Instead, each dyskerin uses a hydrophobic pocket to accommodate either a coil or an alpha helix from the other dyskerin subunit. The asymmetric binding of the two dyskerin molecules therefore blocks the hydrophobic pocket of both subunits, thereby preventing multimerization.

4. It seems interesting that the pseudosubstrate conformation of hTR aids its association with the H/ACA protein complex (Fig. 7). In the case of snoRNPs, however, snoRNAs easily associate with the protein complex in the absence of substrate RNA. Thus, the pseudosubstrate conformation may not be necessary?

We agree with the reviewer that the pseudosubstrate conformation adopted by hTR as observed in our structure may not be necessary for the snoRNPs and scaRNPs. However, in telomerase, there are decreased protein interactions with the 5' hairpin of hTR. This loss is compensated by an hTR-specific increase in protein binding to the 3' hairpin (Egan & Collins, 2010). The pseudosubstrate conformation likely provides additional stabilisation to the interactions between hTR and dyskerin to account for this. Substitutions, mutations or deletions at the 3' pseudosubstrate region reduce or eliminate hTR accumulation (Egan & Collins, 2010, Egan & Collins, 2012). Interestingly, like telomerase, human intron-encoded AluACA RNAs have been shown to have sub-optimal 5' hairpins and a 3' hairpin with additional stabilisation motifs (Ketele et al., RNA Biol. 2016). It remains to be determined whether this pseudosubstrate conformation is only limited to telomerase H/ACA RNP or also extends to the AluACA RNPs.

5. On the other hand, the hTR pseudosubstrate could explain why an artificial substrate matching that sequence is not pseudouridylation unlike those matching other H/ACA RNAs.

We agree with the reviewer's interpretation. The pseudosubstrate RNA region of hTR and the lack of a sequence mimicking the PS1 helix, likely preclude binding of other "substrate" RNAs to hTR. We have now added a sentence to the text (lines 516-517) to include this.

6. Although the model explains the impact of dyskerin DC mutations on the NOP10-dyskerin interaction, it does not address the impact of the R34W NOP10 DC mutation.

The NOP10 R34W lies away from the 3' hydrophobic cleft of dyskerin, and interacts with phosphate backbone of hTR. R34W mutation likely disrupts this interaction and has been described previously (Ghanim et al., 2021).

7. There is no mention, that the long intrinsically disordered N- and C-terminal tails of dyskerin and GAR1 are invisible in the structure. These tails could impact some of the interpretations reached in the manuscript.

We thank the reviewer for pointing this out. We have now added several sentences in the revised manuscript to comment on this point (lines 301-303, 433).

8. Finally, it is not clear what "circle" refers to in Fig. 2b on p.9, line 268?

We thank the reviewer for pointing this out. This refers to the circle in Figure 2b, which shows the position of the hydrophobic cleft. We have now made it more prominent in the figure.

Tom Meier

Reviewer #3 (Remarks to the Author):
Comments to the Author

The discovery of telomerase almost forty years ago highlighted its significance as a potential target for anti-cancer and aging treatments. However, the lack of comprehensive structural information has impeded the progress in developing drugs against telomerase. This research addresses this issue by improving the data processing procedures of an existing dataset, resulting in the determination of a 2.7 Å resolution structure of the H/ACA RNP lobe of human telomerase. This level of resolution enables the identification of intricate molecular interactions within the H/ACA RNP, the mapping of disease-related mutations, and the potential guidance for structure-based drug design. Intriguingly, although not yet confirmed, this study proposes that human telomerase possesses pseudouridylation activity based on the observation of two conformations of the 3' dyskerin thumb loop. However, several questions remain to be answered before this research can be published.

We thank the reviewer for the positive remarks.

1. In methods, please combine the two parts: "Model building and refinement" and "AlphaFold2 prediction".

As suggested by the reviewer, we have now combined the two sections in the revised manuscript (lines 232-260).

2. This study addresses the high resolutions of the two maps of the telomerase H/ACA RNP: semi-closed state and open state. Further evaluation techniques are needed, like applying the Q-score to demonstrate and confirm the resolvability of individual hTR and protein subunits' residues, rather than merely providing an overall average value in the table.

We thank the reviewer for the helpful suggestion. We have now included the Q-scores for each complex in the Supplementary Data 3 and 4. Supplementary Figure 3, Supplementary Figure 4 and Supplementary Figure 11 also show the map and model of individual subunits and residues discussed in the manuscript.

3. Line 253-255, delete it or simply summarize its results.

We have modified the sentence to summarize the results as suggested by the reviewer (line 278-280).

4. To gain insights into the pathogenic mechanisms of mutations, certain functional assays must be conducted. These assays aim to investigate the impact of mutations on factors such as enzyme assembly, activity, and other relevant aspects.

We agree with the reviewer's point (please see also Points 1 and 9 by Reviewer 1). However, the reconstitution approach used to prepare telomerase for biochemical and structural studies in this study and also in the telomerase field in general limits our ability to perform biochemical validation. To reconstitute telomerase, we overexpressed only TERT and hTR in human cells. Therefore, we rely on endogenous H/ACA components, which are present in a vast

cellular excess compared to TERT and hTR, to assemble with the overexpressed TERT and hTR to form telomerase holoenzyme. We also used a hepatitis delta virus (HDV) ribozyme at the 3' end of hTR and thus bypass the cellular 3' end processing of hTR. Under these conditions, performing *in vivo* mutagenesis properly will require endogenous knockout of telomerase components, which is lethal for cells because of their involvement in highly active ribosome and spliceosome biogenesis pathways. Likewise, knockdown of H/ACA RNP components will also affect cell survival. Furthermore, to obtain sufficient telomerase for biochemical characterisation, we transfect a large number of plates of cells. At such scale, knocking down dyskerin would be challenging. In addition, to our current knowledge, telomerase biogenesis and the 3' end processing of hTR are active areas of research and not very well-understood. Therefore, even if we could generate mutants, the tools to dissect the effects of the mutants are well beyond our expertise and outside the scope of this manuscript.

However, to address the reviewer's comment, we performed the best possible experiments that our current experimental setup allows, and have described them in detail below.

Regarding interactions involving dyskerin, we prepared overexpression constructs of Flag-tagged wild-type and mutant dyskerin. We selected 8 disease mutations in dyskerin at residues involved in key dyskerin-dyskerin and dyskerin-hTR interactions described in the manuscript, including I30M, Q31E, Q31K, F36V, K43E, L56S, H68A and T66A/T67A/H68A. We transfected these dyskerin constructs together with the TERT and hTR constructs. As mentioned above, endogenous dyskerin was still present in these experiments. We used Western blot to monitor the expression levels of TERT and Flag-tagged dyskerin and also the level of endogenous dyskerin in the lysate (Supplementary Fig. 7a). Although the same amounts of Flag-tagged dyskerin DNA were used for transfection, the expression levels of all mutant dyskerin were lower than the Flag-tagged wildtype dyskerin (Supplementary Fig. 7a). It is possible that these mutations cause instability in dyskerin. We then performed oligo-based purification on hTR. We observed that the overexpressed Flag-tagged wild-type dyskerin efficiently outcompeted endogenous dyskerin for incorporation into telomerase, evident in the high Flag-dyskerin/total dyskerin ratio in the Western blot (Supplementary Fig. 7b and c). In contrast, all dyskerin mutants competed less efficiently with the endogenous dyskerin for incorporation into telomerase. This is likely caused by the weakened inter-dyskerin or dyskerin-hTR interactions in the dyskerin mutants, validating the interactions we observed in our structures. We performed these experiments in three independent replicates and our observations were consistent across the replicates (Supplementary Fig. 7).

To avoid bypassing 3' end processing of hTR, we used a different hTR expression construct (pBS-U3-hTR-500), in which the HDV ribozyme sequence is replaced with 500 nucleotides of the endogenous 3' UTR of hTR (Fu et al., 2003). To validate the interactions involving nucleotide G450 of hTR, we reconstituted telomerase with G450A, G450C and G450U mutations and performed activity assays (Supplementary Fig. 9). Unlike the experiments included in the submitted manuscript, these assays showed that these mutations resulted in a slight but significant reduction of telomerase activity. In our model, G450 stacks with H68 of the 3' dyskerin and hydrogen bonds with S42 of the 5' dyskerin. The effects are not large because all alternative nucleotides can maintain the stacking interactions with H68 to some extent. The results are consistent across three independent replicates. Mutating G450 to A maintains the purine-H68 stacking while disrupting the hydrogen bond with S42. On the other hand, mutating G450 to a pyrimidine (C or U) reduces the stacking interaction with H68 and disrupts the hydrogen bond with S42. Therefore, the reductions in telomerase activity we observed in G450C and G450U were more pronounced than that in G450A.

We have incorporated these new experiments in the revised manuscript (lines 106-137, lines 316-326 and 368-384).

5.Line 293: a little confusing, need to clarify the direct interaction.

We thank the reviewer for the suggestion. We have now rewritten this sentence to improve clarity (lines 333-336).

6.Line 310, AlphaFold prediction can be done to verify it.

Despite the success of AlphaFold2 in structure prediction using wild-type sequences, we were cautious to use AlphaFold2 for predicting the effects of point mutations. AlphaFold gives the disclaimer that "AlphaFold has not been validated for predicting the effect of mutations." (<https://alphafold.ebi.ac.uk/faq>). This also has been investigated in Pak et al., 2023 (10.1371/journal.pone.0282689), and the authors reached a similar conclusion. However, as suggested by the reviewer, we have conducted AlphaFold2 predictions with the disease mutations as shown in the Figure inserted below. While we observe a change in the overall structure of the HBM fold of dyskerin with the mutants, the prediction confidence is very low.

7.The two parts “Consensus map reveals a non-catalytic semi-closed conformation of the 3' dyskerin thumb loop” and “hTR mimics the substrate-guide RNA duplex of a H/ACA RNP” can be combined to prevent unnecessary duplication.

We understand the concern of the reviewer. However, these two points are quite different. We believe that combining these would make the whole section very lengthy and the two separate conclusions would be less clear and become convoluted.

8.Line 406: a little controversial with the above mentioned “a well-resolved thumb”, and please highlight it in the figure.

We may not have been clear in our text. The heterogeneity was noted during the data processing procedure of the consensus map at an early stage of processing. Upon further classification, the 3' thumb loop is well-resolved in the closed conformation in our consensus map as shown in the inset in Supplementary Figure 3e. We have revised the text to clarify this point (lines 467-468).

9.The authors have postulated the potential presence of pseudouridylation activity in telomerase. Nonetheless, there remains a question regarding the mechanism by which hTR switches with substrate RNA.

We understand the reviewer's concern. This part in the Discussion is pure speculation to open up new thoughts for future studies. However, we have no evidence to this. We have now removed this part of the sentence to avoid any misunderstanding in our speculations.

10.To substantiate the authors' proposal, it is imperative to conduct an in vitro assay to assess the pseudouridylation activity of telomerase.

As mentioned in point 9, this part was purely speculation and we have now removed this part in the Discussion of the main text.

REVIEWERS' COMMENTS

Reviewer #1 (Remarks to the Author):

The authors have sufficiently addressed all my raised concerns, and this manuscript is ready for publication. Thank you.

CJ

Reviewer #2 (Remarks to the Author):

The authors satisfactorily addressed all my concerns and it seems also those of the other reviewers.

Reviewer #3 (Remarks to the Author):

We very much appreciate the detailed responses made to our comments. We now agree that this manuscript is suitable for publication in Nature Communications.

Kaiming Zhang